# Symmetrized Schrödinger Bridge Matching

## Abstract

Schrödinger bridge (SB) has demonstrated numerous applications in probabilistic generative modeling. Finding the solution of probability paths aligns with entropy-regularized optimal transport that employs the Sinkhorn algorithm, which is characterized by performing iterative proportional fitting between marginal densities. This paper argues that the standard training of the SB is prone to exaggerate the amount of learning due to its inherent geometric nature. We leverage a symmetrized variant of Sinkhorn to study more lenient convergence of Schrödinger potentials and prove distinctive theoretical properties of the symmetrization such as linear convergence and monotonic improvements. To this end, we propose a dynamic SB algorithm named Symmetrized Schrödinger Bridge Matching (SSBM). Inspired by score and flow matching models, the concurrent projection scheme of SSBM is conceptualized as matching forward and backward drifts concurrently, constructing a time-symmetric learning objective for the SB model. We empirically validate our SB method by solving classical optimal transportation and model-based stochastic optimal control problems with physical dynamics.

## 1 Introduction

The Schrödinger bridge (SB; Schrödinger, 1932) offers a general formulation for the dynamical evolution of a particle system. The corresponding problem has gained popularity by its connection to the entropy regularized Monge-Kantorovich optimal transport (EOT; Peyré et al., 2019), implying various applications in diverse areas such as image processing, natural language processing, and control systems (Pavon & Wakolbinger, 1991; Léonard, 2012; Caron et al., 2020; Liu et al., 2023; Alvarez-Melis & Jaakkola, 2018; Chen et al., 2022). For its computation, the SB problem is typically solved by the Sinkhorn algorithm (Sinkhorn & Knopp, 1967; Cuturi, 2013), relying on iterative projections between marginals. The algorithm is renowned for the simplicity and the convergence properties inherent to iterative proportional fitting (IPF; Kullback, 1968; Ruschendorf, 1995).

There has been great advancement in synthesizing complex data distributions for deep generative models. Score-based models (Song et al., 2021) seek to find nonlinear functions that transform simple distributions into complex data distributions. These models are characterized by learning the time-reversal process of progressive diffusion starting from data (Sohl-Dickstein et al., 2015), through matching the score function of a stochastic differential equation (SDE). Another line of research involves flow matching (Lipman et al., 2023), which stems from deterministic conditional OT paths between marginals. This is well-described by a continuous vector field of probability ordinary differential equation (ODE), which governs a direct way of translating one distribution to another (Chen et al., 2018). The success of both approaches is supported by nonlinear computational models such as neural networks and corresponding learning schemes for their guidance.

Recent studies have highlighted that SB succeeds in fundamental aspects of score and flow matching models (Liu et al., 2023; Shi et al., 2023). For instance, Learning of SB generally performs score matching where the first training stage of IPF is equivalent to the exact score matching. The projection corresponds to the variational lower bound maximization, or Kullback-Leibler (KL) projection under the Girsanov theorem (Huang et al., 2021). On the other hand, the fluid dynamics formulation of SB generalizes flow-based model by a time-symmetric drift field of probability flow (Nelson, 2001). Despite the strong resemblance, we claim that the direct extension of Sinkhorn only partially embraces the advancement of deep generative models due to the strict geometric constraint of IPF's alternation. In order to efficiently solve the SB problem with a handful of networks at most, we investigate whether there is a more lenient way of training SB models at the algorithmic level.

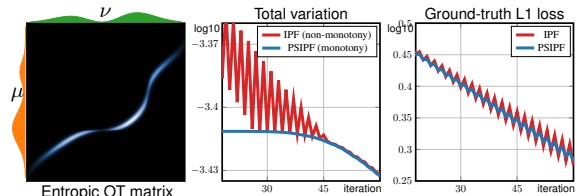

Figure 1: An entropic OT problem and IPFP sequences. In the problem, our PSIPF method (**Algorithm 2**) shows monotonic decrement of temporal variation, and induce more stable learning than standard IPF (**Algorithm 1**).

Table 1: A technical overview. DSB-IPF (De Bortoli et al., 2021) allows dynamical training by drift matching. DSBM-IMF (Shi et al., 2023) preserves marginals during training. SSBM combines these desired properties with stable learning.

|  | Preserving $c \cdot (\mu, \nu)$ | Dynamic | Monotony |
|---|---|---|---|
| DSB | ✗ ($\mu$ or $\nu$) | ✓ | ✗ |
| DSBM | ✓ ($c = 1$) | ✗ | ✗ |
| **SSBM** | ✓ ($c = \kappa$) | ✓ | ✓ |

In this work, we study an alternative learning scheme to the Sinkhorn algorithm. We leverage the concept called *symmetrization* (Kurras, 2015) and propose a novel pseudo-symmetric variant of Sinkhorn for reaching a fair amount of updates at each iteration. We claim that not only does this approach find another way of convergence but also retains distinct properties which help SB training in practice, especially for costly projections involving deep neural networks. For the sake of better understanding, we conducted an actual experiment in Fig. 1. The blue lines in the plots demonstrate that our strategy results in reduced perturbation in both total variation and ground-truth loss.

For the detailed comparison, **Algorithm 1** outlines the discrete Sinkhorn-Knopp algorithm. For a cost matrix $\mathbf{C}$, its objective is to model optimal transport coupling *i.e.* $\pi_* = \mathrm{diag}(\mathbf{u}_*)\mathbf{K}\,\mathrm{diag}(\mathbf{v}_*)$, which represents a coupling between $\mu$ and $\nu$. The consecutive IPF projections are represented in Lines 3 and 4. A discretized version of our approach, called peuedo-symmetric IPF, is demonstrated in **Algorithm 2**. Note that the projection onto coupled $(\tilde{\mathbf{u}}_\ell, \tilde{\mathbf{v}}_\ell)$ occurs in parallel from the current budget $(\mathbf{u}_{\ell-1}, \mathbf{v}_{\ell-1})$. Due to the contraction of projective operations between submanifolds (Bauschke & Borwein, 1994), we adjust the iterates with symmetrical division by the square root of the measure contraction coefficient $\kappa_\ell$. For $n$ dimension, the complexity of each iteration is asymptotically bounded as $\mathcal{O}(n^2)$ for both IPF methods.

---

**Algorithm 1** The Sinkhorn-Knopp algorithm (IPF).

**Input:** a pair $\langle \mu, \nu \rangle$, a cost matrix $\mathbf{C}$, $\lambda \in \mathbb{R}^+$.
1: $\mathbf{u}^{(0)} = \mathbf{1}_\mu$, $\mathbf{K} = \exp(-\mathbf{C}/\lambda)$
2: **for** $n = 1$ **to** $N$ **do**
3:     $\mathbf{v}^{(2n-1)} = \nu \oslash [\mathbf{K}^\mathsf{T}\mathbf{u}^{(2n-2)}]$
4:     $\mathbf{u}^{(2n)} = \mu \oslash [\mathbf{K}\ \mathbf{v}^{(2n-1)}]$
5: **return** $\mathrm{diag}(\mathbf{u}^{(2N)})\mathbf{K}\,\mathrm{diag}(\mathbf{v}^{(2N-1)})$

---

**Algorithm 2** A pseudo-symmetric variant (PSIPF).

1: $\mathbf{u}^{(0)} = \mathbf{1}_\mu$, $\mathbf{v}^{(0)} = \mathbf{1}_\nu$, $\mathbf{K} = \exp(-\mathbf{C}/\lambda)$
2: **for** $\ell = 1$ **to** $L$ **do**
3:     $\tilde{\mathbf{u}}^{(\ell)} = \mu \oslash [\mathbf{K}\ \mathbf{v}^{(\ell-1)}]$
4:     $\tilde{\mathbf{v}}^{(\ell)} = \nu \oslash [\mathbf{K}^\mathsf{T}\mathbf{u}^{(\ell-1)}]$
5:     $\kappa_\ell = \mathrm{sum}[\mathrm{diag}(\tilde{\mathbf{u}}^{(\ell)})\mathbf{K}\,\mathrm{diag}(\tilde{\mathbf{v}}^{(\ell)})]$
6:     $\mathbf{u}^{(\ell)} = \tilde{\mathbf{u}}^{(\ell)}/\sqrt{\kappa_\ell}$, $\mathbf{v}^{(\ell)} = \tilde{\mathbf{v}}^{(\ell)}/\sqrt{\kappa_\ell}$
7: **return** $\mathrm{diag}(\mathbf{u}^{(L)})\mathbf{K}\,\mathrm{diag}(\mathbf{v}^{(L)})$

---

To this end, we propose Symmetrized Schrödinger Bridge Matching (SSBM), a practical algorithm for training Schrödinger bridge, similar to well-established score and flow matching methods. First, we formally state the theoretical benefits of linear convergence and the monotonic improvement in the static SB problem. We then devise our learning algorithm for the dynamic SB problem, which is based on an entropic version of optimal transport in mass-preserving fluid dynamics (Benamou & Brenier, 2000). For the matching objective, we concurrently train both forward and backward models and construct a time-symmetric learning objective for the SB model. As shown in Table 1, one of the key features of our framework is the preservation of both score functions of marginal distributions, which is a similar property appeared in deep Schrödinger bridge matching (DSBM; Shi et al., 2023). However, our approach, unlike DSBM, does not have the constraint of tractable reference measure, which allows us to train a dynamic SB model with arbitrary physical dynamics.

**Our contributions** are three-fold. First, we present a symmetrization scheme, which has theoretically pleasing properties that reduce the instability of neural network training. Second, we devise a matching algorithm that allows easier training. Lastly, we validate SSBM to OT benchmarks and stochastic control problems with physical dynamics and compare it to other SB algorithms.

## 2 SCHRÖDINGER BRIDGE PROBLEM AND SYMMETRIZED SINKHORN

**Schrödinger Bridge Problem (SBP).** Let $(\mathcal{X}, \mu)$ and $(\mathcal{Y}, \nu)$ be Polish spaces. For marginals $(\mu, \nu)$ and the associated path measure $\mathcal{P}(\mu, \nu)$ with a given time interval $[0, T]$, the formal de-

scription of SBP is to find the KL projection $\mathbb{P}^{\text{SB}} := \inf_{\mathbb{P} \in \mathcal{P}(\mu,\nu)} H(\mathbb{P}|\mathbb{Q})$ where $\mathbb{Q} \in \mathcal{P}(\mu,\nu)$ is a reference measure. Let $\Pi(\mu,\nu)$ be the set of couplings and $c$ be a continuous cost function. By the disintegration of measures (Léonard, 2014), the relative entropy of SBP yields the chain rule

$$H(\mathbb{P}|\mathbb{Q}) = H(\pi|\mathcal{G}) + \iint_{\mathcal{X} \times \mathcal{Y}} H(\mathbb{P}_\pi|\mathbb{Q}_\mathcal{G}) \, \mathrm{d}\pi(x,y), \tag{1}$$

where $\otimes$ is the product of measures and $\mathcal{G}$ denotes a Gibbs measure $\mathrm{d}\mathcal{G} \propto e^{-c_\lambda} \mathrm{d}(\mu \otimes \nu)$ for $c_\lambda := c/\lambda$ with $\lambda \in \mathbb{R}^+$. Enforcing the conditional probabilities $\mathbb{P}_\pi = \mathbb{Q}_\mathcal{G}$ yields the problem:

$$\inf_{\pi \in \Pi(\mu,\nu)} H(\pi|\mathcal{G}) = \iint_{\mathcal{X} \times \mathcal{Y}} c(x,y) \, \mathrm{d}\pi(x,y) + \lambda H(\pi|\mu \otimes \nu) + \text{const.} \tag{2}$$

Therefore, the static SBP is equivalent to the standard EOT with a $\lambda$-regularizer. This relationship allows us to consider the problem as finding an optimal EOT plan $\pi_* \in \Pi(\mu,\nu)$ (Peyré et al., 2019).

**Sinkhorn Algorithm.** The constrained optimization (2) naturally yields the strong duality. In particular, consider the *Schrödinger potentials* $(\varphi_*, \psi_*)$, which constitute $\pi_*$ with the Radon-Nikodym derivative: $\mathrm{d}\pi_* = e^{\varphi_* \oplus \psi_* - c_\lambda} \mathrm{d}(\mu \otimes \nu)$. Then, the following statement holds for the potentials.

**Lemma 2.1** (Duality of SBP; Theorem 3.2 of Nutz, 2021). *Assume the existence of Schrödinger bridge $\pi_* \in \Pi(\mu,\nu)$ and corresponding Schrödinger potentials $(\varphi,\psi) \in L^1(\mu) \times L^1(\nu)$. Then,*

$$\min_{\pi \in \Pi(\mu,\nu)} H(\pi|\mathcal{G}) = \sup_{\varphi,\psi} F(\varphi,\psi), \quad F(\varphi,\psi) := \mu(\varphi) + \nu(\psi) - \iint_{\mathcal{X} \times \mathcal{Y}} e^{\varphi \oplus \psi} \, \mathrm{d}\mathcal{G} + 1, \tag{3}$$

*where $\oplus$ indicates the direct sum of two potentials and $\mu(\varphi) := \int_\mathcal{X} \varphi \, \mathrm{d}\mu$ and $\nu(\psi) := \int_\mathcal{Y} \phi \, \mathrm{d}\nu$.*

From a geometric perspective, the Sinkhorn updates are characterized by differentiating the dual functional $F$. As a result, the algorithm performs alternating projections (Nutz & Wiesel, 2023):

$$\psi_{2n-1}(y) = -\log \int_\mathcal{X} e^{\varphi_{2n}(x) - c_\lambda(x,y)} \mu(\mathrm{d}x), \quad \varphi_{2n}(x) = -\log \int_\mathcal{Y} e^{\psi_{2n-1}(y) - c_\lambda(x,y)} \nu(\mathrm{d}y), \tag{4}$$

for all $(x,y) \in \mathcal{X} \times \mathcal{Y}$, and these operations are essentially linear in terms of exponential. The estimation of the coupling from the current budget naturally split into two versions:

$$\mathrm{d}\pi_{2n-1} = e^{\varphi_{2n-2} \oplus \psi_{2n-1} - c_\lambda} \mathrm{d}(\mu \otimes \nu), \qquad \mathrm{d}\pi_{2n} = e^{\varphi_{2n} \oplus \psi_{2n-1} - c_\lambda} \mathrm{d}(\mu \otimes \nu). \tag{5}$$

Note that the acquisition of marginals is also splitted; $\mu$ can be acquired with the first marginal of $\pi(\varphi_{2n}, \psi_{2n-1})$, and $\nu$ with the second marginal of $\pi(\varphi_{2n-2}, \psi_{2n-1})$ by its alternating nature.

**A Symetrization Proposal.** In this work, we propose the following symmetrization framework for SBP which is the direct extension of Algorithm 2. The procedure is composed of two stages. First, the Schrödinger potentials are concurrently updated with intermediate representations:

$$\tilde{\varphi}_\ell(x) = -\log \int_\mathcal{Y} e^{\psi_{\ell-1}(y) - c_\lambda(x,y)} \nu(\mathrm{d}y), \quad \tilde{\psi}_\ell(y) = -\log \int_\mathcal{X} e^{\varphi_{\ell-1}(x) - c_\lambda(x,y)} \mu(\mathrm{d}x). \tag{6}$$

Unlike alternating update in Eq. (5), it is evident that the concurrent operation in Eq. (6) does not satisfy the constraint of $\Pi(\mu,\nu)$; thus, $(\tilde{\varphi}_\ell, \tilde{\psi}_\ell)$ are not associated as potentials. Hence, one can subsequently recover the constraint by equally subtracting a certain amount:

$$\varphi_\ell(x) = \tilde{\varphi}_\ell(x) - \log\sqrt{\kappa_\ell}, \qquad \psi_\ell(y) = \tilde{\psi}_\ell(y) - \log\sqrt{\kappa_\ell}, \tag{7}$$

where $\kappa_\ell$ denotes measure contraction $\kappa_\ell := \iint_{\mathcal{X} \times \mathcal{Y}} e^{\tilde{\varphi}_\ell \oplus \tilde{\psi}_\ell - c_\lambda} \mathrm{d}(\mu \otimes \nu)$. Applying the projection in parallel, the algorithm seeks to recover both marginals involving the scaling factor $\sqrt{\kappa_\ell}$.

*Remark* 2.2. For $\{\pi_\ell\}_{\ell \in \mathbb{N}}$, $\int_\mathcal{Y} e^{\varphi_\ell \oplus \psi_{\ell-1} - c_\lambda} \mathrm{d}\nu = \mu/\sqrt{\kappa_\ell}$ and $\int_\mathcal{X} e^{\varphi_{\ell-1} \oplus \psi_\ell - c_\lambda} \mathrm{d}\mu = \nu/\sqrt{\kappa_\ell}$.

Compared to the standard Sinkhorn, the estimation of coupling from current budget writes in a singular form $\mathrm{d}\pi_\ell = e^{\varphi_\ell \oplus \psi_\ell - c_\lambda} \mathrm{d}(\mu \otimes \nu)$. This work refers to the procedure as *symmetrized* Sinkhorn.

## 3 THEORETICAL ANALYSES ON SYMMETRIZED SINKHORN

This section analyzes the associated sequences *i.e.* $\{\varphi_\ell\}_{\ell \in \mathbb{N}}$, $\{\psi_\ell\}_{\ell \in \mathbb{N}}$, and $\{\pi_\ell\}_{\ell \in \mathbb{N}}$ in their convergence and theoretical properties. Assume that projections occur in finite-dimensional spaces and the boundedness of the cost function. Under these assumptions, we show the linear convergence of symmetrized Sinkhorn for the dual functional $F$; thus, our method leads the potentials to a unique fixed point of convergence with gradual improvements.

**Proposition 3.1** (Linear convergence). *Suppose a bounded cost, and let $(\mathcal{X}, \mathcal{F}_1, \mu)$ and $(\mathcal{Y}, \mathcal{F}_2, \nu)$ be the probability spaces. The sequence of symmeterized Sinkhorn iterates $(\varphi_\ell, \psi_\ell)$ converges strongly in $L^p(\mathcal{X}, \mathcal{F}_1, \mu) \times L^p(\mathcal{Y}, \mathcal{F}_2, \nu)$ for $p \in [1, \infty]$. Upon the existence of the solution $\pi_*$,*

$$F(\varphi_*, \psi_*) - F(\varphi_\ell, \psi_\ell) \leq k^\ell \big(F(\varphi_*, \psi_*) - F(\varphi_0, \psi_0)\big), \qquad \ell \in \mathbb{N} \tag{8}$$

*holds, where $k = 1 - e^{-22\|c_\lambda\|_\infty} \in (0, 1)$ and $(\varphi_*, \psi_*)$ are the optimal potentials for $F$.*

Notice that our analysis achieves a slightly tighter contraction than $(1 - e^{-24\|c_\lambda\|_\infty})$ of centered Sinkhorn (Carlier, 2022), and the reason of the difference is mainly due to the fact that we put more the number of projections per iteration. Therefore, this result further suggests that increasing the number of projections helps choosing fair amount of SB training.

Meanwhile, the Birkhoff-Bushell theorem (Birkhoff, 1957; Bushell, 1973) predicts measure contraction property such that $\log \kappa_\ell \leq 0$. Since the suboptimality gap gradually gets minimal by Eq. (8), $\log \kappa_\ell$ are monotone increasing to 0. Using this property, we present the monotony of the algorithm in terms of relative entropy for sufficiently large iterations. It is is related to the well-known monotony of the Sinkhorn iterates $\{H(\pi_*|\pi_{2n})\}_{n\geq 0}$ and $\{H(\pi_*|\pi_{2n-1})\}_{n\geq 0}$ (Nutz, 2021). However, the drawback is that $H(\pi_*|\pi_n) \leq H(\pi_*|\pi_{n+1})$ does not hold; thus the inconsistency might persist especially when the IPF projections are estimated with a finite number of neural networks.

**Proposition 3.2** (Monotony of symmetrized Sinkhorn). *Suppose that the EOT coupling $\pi_*$ exists. For sufficiently large iteration $\ell$, the relative entropy between coupling monotonically decreases for the iterates drawn from the symmetrized Sinkhorn, denoting $H(\pi_*|\pi_\ell) \leq_\ell H(\pi_*|\pi_{\ell-1})$.*

In a computational context, our algorithm is also stable with discrete measures (*i.e.*, mini-batch learning), as it inherits geometric convergence properties from Sinkhorn (see Theorem 2.1 of Nutz & Wiesel, 2023). To summarize, we have found that our algorithm brings theoretically pleasing properties, namely, linear convergence and monotonic improvement. Compared to established probabilistic generative methods, training of an SB model has been criticized for its complexity and instability in finding solutions (Liu et al., 2023). Our hypothesis is that the benefits from the symmetrization holds for general SB problems. We claim that our symmetrization scheme leads to more stable results than standard Sinkhorn, especially when solving SBP relies on finite models, and the correlation between subsequent iterations is considerable.

## 4 SYMMETRIZED SCHRÖDINGER BRIDGE MATCHING

Based on the theoretical analyses, we aim to apply the aforementioned symmetrization framework to general controllable dynamics for robust training of models. We propose a deep dynamic SB algorithm, which we call SSBM. To computationally model the notion of concurrent projection, we utilize the essential technique from matching algorithms (De Bortoli et al., 2021; Lipman et al., 2023) to provide symmetric targets. This section sets the regularization coefficient $\lambda$ to 1.

### 4.1 DYNAMIC CONTROL FORMULATION OF SB

Suppose that stochastic processes control the path measures $\mathbb{P}^\mu$ and $\mathbb{P}^\nu$ starting from $\mu$ and $\nu$, respectively. If these two measures form SB between the marginals, the solution is represented with time-varying potentials $(\Psi, \widehat{\Psi}) \in C^{1,2}([0, T], \mathbb{R}^n)$ which construct coupled SDEs:

$$dX_t = \big[f(t, X_t) + gg^\mathsf{T}(t, X_t)\nabla\log\Psi(t, X_t)\big]\,dt + g(t, X_t)\,dW_t, \qquad X_0 \sim \mu, \tag{9a}$$

$$d\bar{X}_s = \big[-f(s, \bar{X}_s) + gg^\mathsf{T}(s, \bar{X}_s)\nabla\log\widehat{\Psi}(s, \bar{X}_s)\big]\,ds + g(s, \bar{X}_s)\,dW_s, \quad \bar{X}_0 \sim \nu, \tag{9b}$$

where $f$ and $g$ denote base drift and diffusion function given by the environment. In the SDEs, $X_t$ evolves with the "forward" equation (9a), and $\bar{X}_s$ also evolves, but with the "reversed" time coordinate $s := T - t$. It is well known that $\Psi$ and $\widehat{\Psi}$ satisfy the partial differential equation (PDE):

$$\begin{cases} \frac{\partial\Psi(t,x)}{\partial t} = -\nabla\Psi^\mathsf{T} f - \frac{1}{2}\operatorname{Tr}(gg^\mathsf{T}\nabla^2\Psi) \\ \frac{\partial\widehat{\Psi}(t,x)}{\partial t} = -\nabla\cdot\big(\widehat{\Psi}f\big) + \frac{1}{2}\nabla^2\cdot(gg^\mathsf{T}\widehat{\Psi}) \end{cases} \text{such that} \quad \begin{aligned} \Psi(0,\cdot)\widehat{\Psi}(0,\cdot) &= \mu, \\ \Psi(T,\cdot)\widehat{\Psi}(T,\cdot) &= \nu, \end{aligned} \tag{10}$$

where the operator $\nabla^2$ and $\nabla^2\cdot$ denotes a shorthand notation for the Hessian and a nested divergence operator for matrix functions. The PDE (10) suggests that $(\Psi, \widehat{\Psi})$ formulates the solutions

to *minimum control* (or entropy-regularized) optimization problem (Bensoussan et al., 2013) while preserving density. Using nonlinear Feynman-Kac (FK) lemma (Han et al., 2018; Pereira et al., 2020), SB studies have presented a deep neural network parameterization according to the forward-backward SDE (SB-FBSDE; Chen et al., 2022; Liu et al., 2022a), where we delineate the detailed derivation to Appendix B. Based on the SB-FBSDE theory, training of one of SDE is based on the backward trajectories sampled from the reversed counterpart, maximizing the likelihood of the reversed path measure.

## 4.2 Time-Symmetric Approach to Dynamical SB Problems

To construct the learning target for both forward and backward SDEs, we utilize an optimal transport formulation in mass-preserving fluid dynamics. Suppose a drift field $v : [0, T] \times \mathbb{R}^d \to \mathbb{R}^d$ and a corresponding probability density $\rho(t, \cdot) \in \mathcal{P}(\mathbb{R}^d)$ where $\mathcal{P}(\mathbb{R}^d)$ is the set of probability measures on $\mathbb{R}^d$. Using the Nelson's duality (Nelson, 2001), we define *time-symmetric* current drift $v_t(x) := \frac{1}{2}[f^+(t, x) - f^-(s, \bar{x})]$ where drifts $f^+(t, x)$ and $f^-(s, \bar{x})$ drawn from the FBSDE. For a transport cost function $c(x, y) = \frac{1}{2}\|x - y\|^2$, an entropic analogue of the Benamou-Brenier formula (Benamou & Brenier, 2000; Gigli & Tamanini, 2020), or *the time-symmetric dynamical SBP* writes

$$H(\mathbb{P}|\mathbb{Q}) = \inf_{(v_t, \rho_t)} \left\{ \int_0^T \int_{\mathbb{R}^d} \left( \underbrace{\frac{1}{2}\|(v_t - f_t)(x)\|^2}_{\text{kinetic energy}} + \underbrace{\frac{1}{8}\|\nabla \log \rho_t(x)\|_{gg^\intercal}^2}_{\text{Fisher information}} \right) \rho_t(x) \mathrm{d}x \mathrm{d}t \,\middle|\, \frac{\partial \rho}{\partial t} + \nabla \cdot (v\rho) = 0 \right\}.$$

(11)

The objective encodes the kinetic energy endowed with a geometry incurred by the Fisher information metric, and the condition on the righthand side is called the *continuity* equation which states the conservation of the probability density. Under mild conditions, Eqs. (1) and (11) are equivalent in terms that the cost of energy in the space of information geometry models the EOT problem.

Just like other SB representations, the optimality is unique, satisfying a Hamilton-Jacobi equation (HJE). In the following proposition, we present the HJE with a function $\Phi$ defined with $(\Psi, \widehat{\Psi})$.

**Proposition 4.1.** *Suppose a function $\Phi \in C^{1,1}([0, T], \mathbb{R}^n)$ and let $f, g$ satisfy growth and Lipchitz conditions. The vector field $v_t(x) := f_t(x) + gg^\intercal \nabla \Phi(t, x)$ corresponds to the solution of Eq.* (11) *if*

$$\frac{\partial \Phi(t, x)}{\partial t} + v_t \cdot \nabla \Phi(t, x) = \frac{1}{4}\|\nabla \log \Psi(t, x)\|_{gg^\intercal}^2 + \frac{1}{4}\|\nabla \log \widehat{\Psi}(s, \bar{x})\|_{gg^\intercal}^2$$

$$\Phi(t, x) := \frac{1}{2}\{\log \Psi(t, x) - \log \widehat{\Psi}(s, \bar{x})\}, \quad s := T - t,$$

(12)

*where the potentials $(\Psi, \widehat{\Psi})$ satisfy the PDE (10).*

Due to the uniqueness of SDE solutions, Eqs. (10) and (12) predict the identical SB structure. In quantum mechanics, $\mathbf{j} = v\rho = \frac{1}{2}(\widehat{\Psi}\nabla\Psi - \Psi\nabla\widehat{\Psi})$ is often called as *probability flux* (Paul & Baschnagel, 1999; Chen et al., 2017), making a concise way of describing the continuity $\partial_t \rho_t + \nabla \cdot \mathbf{j} = 0$. In this context, we can understand the relationship between $(v, \rho)$ and $(\Psi, \widehat{\Psi})$ as two equivalent representations of EOT for a probability path along $(\mu, \nu)$. Since the HJE (12) have the symmetric property, where $(\Psi, \widehat{\Psi})$ both involved regardless of path distributions $\mathbb{P}^\mu$ and $\mathbb{P}^\nu$, we propose to consider the current vector field $v_t$ as a symmetrized learning target for achieving the SB optimality.

## 4.3 Iterative Probability Flux Matching

Under the Girsanov theorem (Øksendal, 2003), maximizing log-likelihoods by matching drifts corresponds to KL projections for another path measures; consequently, this has inspired consecutive SB methods in previous studies (**Algorithm 3**; De Bortoli et al., 2021; Vargas et al., 2021).

**Proposition 4.2** (Girsanov theorem). *For two drifts $f^+$ and $f^-$ from $\mathbb{P}^\mu \in \mathcal{P}(\mu, \cdot)$ and $\mathbb{P}^\nu \in \mathcal{P}(\cdot, \nu)$, define respective probability densities as $(\rho^+, \rho^-)$ and time-reversal drifts $(\gamma^+, \gamma^-)$. Then,*

$$H(\mathbb{P}^\mu|\mathbb{P}^\nu) = \frac{1}{2}\int_0^T \mathbb{E}_{x \sim \rho^+(t, \cdot)}\|(f^+ - \gamma^-)(t, x)\|_{gg^\intercal}^2 \, \mathrm{d}t$$

(13a)

$$H(\mathbb{P}^\nu|\mathbb{P}^\mu) = \frac{1}{2}\int_0^T \mathbb{E}_{x \sim \rho^-(s, \cdot)}\|(f^- - \gamma^+)(s, \bar{x})\|_{gg^\intercal}^2 \, \mathrm{d}s$$

(13b)

*where $H(\cdot|\cdot)$ denotes relative entropy between two path measures.*

| **Algorithm 3** Schrödinger bridge matching (SBM). | **Algorithm 4** Symmetrized SB matching (SSBM). |
|---|---|
| **Input:** $\mu, \nu, \mathbb{P}_0^\mu$ | **Input:** $\mu, \nu, \mathbb{P}_0^\mu, \mathbb{P}_0^\nu$ |
| 1: $n = 1$ | |
| 2: **repeat**      *# consecutive KL projection.* | 2: **repeat**      *# concurrent KL projection.* |
| 3:    $\mathbb{P}_{2n-1}^\nu = \arginf_{\mathbb{P}^\nu \in \mathcal{P}(\cdot, \nu)} H(\mathbb{P}^\nu \| \mathbb{P}_{2n-2}^\mu)$ | 3:    $\tilde{\mathbb{P}}_\ell^\mu = \arginf_{\mathbb{P}^\mu \in \mathcal{P}(\mu, \cdot)} H(\mathbb{P}^\mu \| \mathbb{P}_{\ell-1}^\nu)$ |
| 4:    $\mathbb{P}_{2n}^\mu = \arginf_{\mathbb{P}^\mu \in \mathcal{P}(\mu, \cdot)} H(\mathbb{P}^\mu \| \mathbb{P}_{2n-1}^\nu)$ | 4:    $\tilde{\mathbb{P}}_\ell^\nu = \arginf_{\mathbb{P}^\nu \in \mathcal{P}(\cdot, \nu)} H(\mathbb{P}^\nu \| \mathbb{P}_{\ell-1}^\mu)$ |
| 5:    $n := n + 1$ | 5:    Obtain $\mathbf{j}^\ell$ using $\langle \tilde{\mathbb{P}}_\ell^\mu, \tilde{\mathbb{P}}_\ell^\nu \rangle$ via HJE (15). |
| 6: **until** convergence; | 6:    Update $\langle \mathbb{P}_\ell, \mathbb{P}_\ell \rangle$ using $\mathbf{j}^\ell$ via $\mathcal{L}_{\text{FSDE}}$ and $\mathcal{L}_{\text{BSDE}}$. |
| 7: **return** $\mathbb{P}_*^\mu, \mathbb{P}_*^\nu$. | 7:    $\ell := \ell + 1$ |
| | 8: **until** convergence; **return** $\mathbb{P}_*^\mu, \mathbb{P}_*^\nu$. |

The theorem suggests that one way to achieve KL projection between path measures is by matching drifts with the time reversal drifts of $(\gamma^+, \gamma^-)$. Following DSB and CFM (De Bortoli et al., 2021; Lipman et al., 2023), we train two target drifts $(\tilde{f}_\ell^+, \tilde{f}_\ell^-)$ with conditional drift matching (CDM) loss:

$$\mathcal{L}_{\text{CDM}}^\pm(\ell) = \mathbb{E}_{t, q(x_t) p_{\ell-1}^\mp(x'|x_t)} \big\| \tilde{f}_\ell^\pm(x') - (x - x')/\varepsilon \big\|_{gg^\mathsf{T}(t \mp \varepsilon, x')}^2, \tag{14}$$

where $\pm$ and $\mp$ indicates consideration of signs regarding its timelines ($+$ and $-$), and $p^\pm$ is a discrete Markovian kernel of $\rho_\ell^\pm$ for a small time interval $\varepsilon$. For instance, we can sample data using the Euler-Maruyama integration. If the distribution $q(\cdot)$ covers the desired support set, the relative entropies Eq. (13) and conditional matching loss (14) offer identical gradients to the target networks.

In order to model the dynamic version of symmetrized Sinkhorn, we need a learning method for SDE drifts $f_\ell^\pm$ that preserves probability density along $(\mu, \nu)$. Therefore, we utilize Proposition 4.1 and define the estimated target current drift $v_\ell(t, x) := 1/2[\tilde{f}_\ell^+(t, x) - \tilde{f}_\ell^-(\hat{t}, x)]$ and nonlinear FK transformations $Y_\ell \approx \log \Psi$ and $\hat{Y}_\ell \approx \log \hat{\Psi}$. Hence, we propose the following loss function:

$$\mathcal{L}_\Phi(\ell) = \mathbb{E}_{t,x} \left| \frac{\partial \Phi_\ell}{\partial t} + v_\ell \cdot \nabla \Phi_\ell - \frac{1}{4} \big\| \nabla Y_\ell \big\|_{gg^\mathsf{T}}^2 - \frac{1}{4} \big\| \nabla Y_\ell \big\|_{gg^\mathsf{T}}^2 \right|,$$
$$\Phi_\ell(t, x) := 1/2 \left( Y_\ell(t, x) - \hat{Y}_\ell(s, \bar{x}) \right), \quad s = T - t \tag{15}$$

We also keep the marginal score consistency $\nabla Y_\ell(0, \cdot) + \nabla \hat{Y}_{\ell-1}(0, \cdot) = \nabla \log \mu$ and $\nabla Y_{\ell-1}(T, \cdot) + \nabla \hat{Y}_\ell(T, \cdot) = \nabla \log \nu$ with an auxiliary loss using score-based methods. Consequently, we can achieve the SB model $(Y_\ell, \hat{Y}_\ell)$ that ① traverses with the target vector field $v_\ell$ with density preservation (7), and also ② preserves marginal score functions for both sides (Remark 2.2). The updates are uniquely defined for every iteration.

Finally, the obtained $Y_\ell$ and $\hat{Y}_\ell$ are used to train drifts, the forward drift is trained with the following loss functions achieving maximum likelihood estimation for $\mathcal{P}(\mu, \cdot)$ and $\mathcal{P}(\cdot, \nu)$.

$$\mathcal{L}_{\text{FSDE}}(\ell) = \mathbb{E}_{t,x} \big| f_\ell^+ - \{ \ f(t, x) + g^\mathsf{T} \nabla Y_\ell(t, x) \} \big| \tag{16a}$$

$$\mathcal{L}_{\text{BSDE}}(\ell) = \mathbb{E}_{s,\bar{x}} \big| f_\ell^- - \{ -f(s, \bar{x}) + g^\mathsf{T} \nabla \hat{Y}_\ell(s, \bar{x}) \} \big| \tag{16b}$$

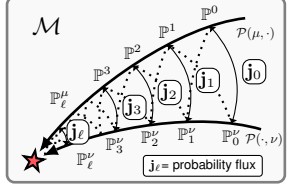

**Algorithm 4** summarizes the SSBM procedure. Notice that we make abstraction for some of the details by using the notion of path measures and probability flux, which have equivalent meanings to training with the proposed loss functions. This abstraction also allows us to understand Fig. 2, which illustrates SSBM, symmetric learning toward an optimum by matching probability flux $\mathbf{j}_\ell = v_\ell \rho_\ell$. We leave more algorithmic details in the appendix.

Figure 2: A schematic illustration. Two projection (dashed lines) operates between path measures. SSBM identifies a distinct updates for target flux $\mathbf{j}_\ell$ within the information geometry $\mathcal{M}$.

## 5 RELATED WORKS

We are interested in one fundamental aspect of Schrödinger bridge (Schrödinger, 1931; 1932), specifically its equivalence with EOT structures (Peyré et al., 2019; Nutz, 2021). In machine learning, there has been progress in training SB with nonlinear networks with Sinkhorn algorithm (Vargas et al., 2021; De Bortoli et al., 2021; Chen et al., 2022). Recently, the general convergence of

Sinkhorn for various conditions has been extensively studied (Peyré et al., 2019; Nutz & Wiesel, 2023; Deng et al., 2023; Chen et al., 2023). As a symmetric counterpart, we propose symmetrized Sinkhorn, which extends PSIPF (Kurras, 2015) to SB problems while retaining theoretically pleasing features of PSIPF, which are advancements from the analyses of (Carlier, 2022; Nutz, 2021).

Score-based methods have exhibited exceptional image generation for diffusion models (Ho et al., 2020; Song et al., 2021). From a perspective of variational methods, such score-matching algorithms can be considered as iteratively elevating a lower bound of maximum likelihood estimation through backward stochastic integration (Huang et al., 2021). On the other hand, the flow matching algorithms (Lipman et al., 2023) model vector fields of conditional flow, which often leads to efficient regression model for static OT. It has been verified that the SB is aligned with both score and flow matching (Liu et al., 2023; Shi et al., 2023). By leveraging iterative minimization of KL divergence between path measures (Øksendal, 2003; Vargas, 2021), training SB models have been more inclined toward score matching. Nelson (2001) displayed a time-symmetric configuration of diffusion bridge, uncovering the duality between stochastic process and vector field of mass flow.

The optimal control formalization of SB (Pavon & Wakolbinger, 1991; Léonard, 2012) put each control agent in the symmetrical game with their respective timeline; the goal is to model controlled SDEs with minimum control (Van Handel, 2007). The Hamilton-Jacobi equation offers a dynamic programming approach through a well-formulated PDE (Kirk, 1970; Zavidovique, 2020). Using the SB-FBSDE theory, a multi-step temporal difference method has been proposed Liu et al. (2022a) via backward stochastic integration of the BSDEs (Bellman, 1954; Sutton & Barto, 2018). However, recent work shows that treating BSDE as updates could struggle to find convergent solutions due to the stochastic cost variance (Andersson et al., 2023). The time-symmetrical HJE has been theoretically studied (Chen et al., 2016; Gigli & Tamanini, 2020), which models entropy regularized Benamou-Brenier formula (Benamou & Brenier, 2000) for EOT in mass-preserving fluid dynamics.

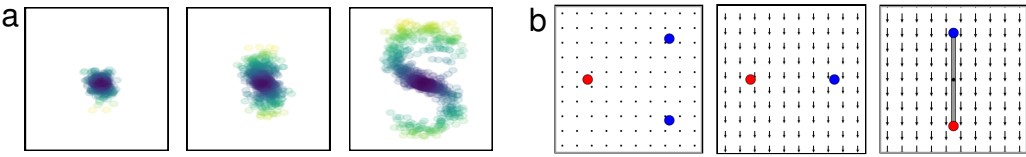

Figure 3: An overview of the experiment. (a) Classical OT experiment in Euclidean space. We also varied the marginal distributions and dimensional to measure stability and solubility. (b) Stochastic optimal control experiments with various physical dynamics. The control is represented as an external force, and there are other forces in the environment, such as drag and gravitational force.

## 6 EXPERIMENTAL RESULTS

We validated our SSBM on two classes of SB problems, including classical OT problem and general optimal control problem (Fig. 3). The goal of the OT experiment was to validate the stability of the SSBM approach in comparison to prior methods under diverse configurations. Also, we subjected the optimal control variant of SSBM to validation within second-order dynamics as the underlying physical system. These systems have served as the foundational physical framework and have been a subject of classical control studies (Abraham & Marsden, 2008). We parameterized the functions with fully connected deep neural networks. OT networks adopted sinusoidal time embeddings and were trained with AdamW. We set $\lambda = 1$ and SDEs were solved with the Euler-Maruyama method.

**2D OT Experiments.** We first show our proposed method achieved competitive OT performance in 2D problems. We compared our method SSBM with DSB (De Bortoli et al., 2021), DSBM-IMF (Shi et al., 2023), Rectified Flow (RF; Liu et al., 2022b), SB-CFM (Tong et al., 2023). RF uses an iterative flow matching procedure in order to improve the straightness of the flow, and SB-CFM utilizes batch-wise Sinkhorn solvers to define an approximate SB static coupling. Since our algorithm used the conditional drift matching loss Eq. (14) to construct a target current drift $v_\ell$, the algorithmic improvements compared to standard Sinkhorn are closely related to comparison with DBM. Table 2 shows the OT experiment among four different types of marginal distributions. In total variations, SSBM achieved the five best results for eight configurations. In path relative

Table 2: OT performance evaluated using path relative entropies and marginal total variations across four 2D experiments (5 runs). The best outcomes among these are highlighted in bold.

| Dataset | Forward Path Relative Entropy | | | | Backward Path Relative Entropy | | | |
|---|---|---|---|---|---|---|---|---|
| | gaussian | multimodal | s-curve | moon | gaussian | multimodal | s-curve | moon |
| DSB | 411.872±633.015 | 22.432±13.328 | 19.481±11.717 | 6.398±1.599 | 33.097±17.383 | 120.255±234.005 | 19.089±10.305 | 6.369±1.755 |
| DSBM | 8.936±0.294 | **3.864±0.276** | 6.866±0.304 | 3.518±0.190 | 8.942±0.291 | **3.942±0.256** | 6.939±0.203 | 3.469±0.193 |
| SB-CFM | **8.877±0.310** | 4.067±0.360 | **6.829±0.175** | **3.342±0.150** | **8.893±0.324** | 4.119±0.287 | **6.912±0.042** | **3.341±0.120** |
| **SSBM** | 9.560±0.540 | 5.874±0.389 | 7.626±1.418 | 5.718±0.970 | 9.593±0.623 | 5.629±0.294 | 9.544±0.591 | 5.075±0.811 |
| Dataset | Temporal Variation ($\mu$) | | | | Temporal Variation ($\nu$) | | | |
| | gaussian | multimoidal | s-curve | moon | gaussian | multimoidal | s-curve | moon |
| DSB | 2.903±0.797 | 8.893±13.599 | 0.930±0.009 | 2.352±0.215 | 13.140±15.666 | 5.105±2.846 | 3.997±0.269 | 3.141±0.079 |
| DSBM | 2.301±0.037 | 2.276±0.039 | 0.494±0.016 | 2.280±0.024 | 2.260±0.040 | 3.383±0.060 | 3.651±0.029 | 3.216±0.029 |
| RF | 2.802±0.117 | 2.384±0.034 | 1.417±0.037 | 2.374±0.017 | 2.345±0.046 | 3.241±0.032 | **2.870±0.036** | **3.071±0.030** |
| SB-CFM | 2.259±0.032 | 2.226±0.058 | **0.452±0.017** | 2.198±0.043 | 2.285±0.049 | 3.385±0.055 | 3.633±0.037 | 3.192±0.024 |
| **SSBM** | **1.906±0.070** | **1.911±0.064** | 0.493±0.008 | **1.877±0.033** | **1.928±0.073** | **3.130±0.032** | 3.091±0.037 | 3.147±0.089 |

entropy, DSBM and SB-CFM showed remarkable performance. This is due to the usage of reference measures and static SB solvers, which often leads to stabilized energy levels in the static settings. This showed that our SSBM method achieved stable OT results, which are aligned with our theory.

**High-Dimensional Gaussian.** Next, we conducted large-scale Gaussian OT experiment that appeared in (De Bortoli et al., 2021) with $d \in \{1, 20, 50\}$ to verify the scalability of our method. In Table 3, we quantified the accuracy, which shows that our SSBM showed better results for both marginals, and the gap between SSBM and other algorithms increased with the dimension. The results are closely related to the analysis of SB for Gaussian measures from (Bunne et al., 2023), that the entropic Benamou-Brenier equation dictates *how* the mass should be transported globally, rather than focusing on the quantity of the mass. Thus, we conclude that the symmetrization incurs scalability for large data.

Table 3: Average total variation of OT in multi-dimensional Gaussian distributions.

| TV($\mu$) | $d = 1$ | $d = 20$ | $d = 50$ |
|---|---|---|---|
| DSB | 1.458±0.450 | 27.119±0.472 | 70.927±2.200 |
| DSBM | 1.138±0.027 | 23.742±0.169 | 106.980±1.104 |
| RF | 1.227±0.004 | 22.935±0.062 | 57.592±0.173 |
| SB-CFM | 1.131±0.065 | 22.618±0.024 | 64.792±0.756 |
| **SSBM** | **0.966±0.017** | **19.955±1.516** | **48.646±3.754** |
| TV ($\nu$) | $d = 1$ | $d = 20$ | $d = 50$ |
| DSB | 10.327±13.274 | 26.934±0.769 | 70.690±1.805 |
| DSBM | 1.121±0.013 | 23.601±0.116 | 106.254±0.818 |
| RF | 1.078±0.031 | 22.942±0.063 | 57.598±0.179 |
| SB-CFM | 1.091±0.023 | 22.618±0.035 | 64.797±0.847 |
| **SSBM** | **0.931±0.029** | **19.646±1.483** | **44.995±0.994** |

**2D Physical Mass Control.** We considered environments characterized by point mass dynamics operating under second-order principles. In these scenarios, two agents are initially located at distant positions at a steady state. The objective was to establish dynamic SB between starting and goal positions, where we considered stochastic control by generating force. The simulations focused on two distinct settings. In the *Branching environment*, there are one initial and two goal positions. The environments consist of drag forces proportional to the velocities, which take the kinetic energy, and eventually, the systems halt without external control. In the *Gravitational environment*, there is a constant gravitation force in the $y$-axis; thus, each control agent needs to control against gravity.

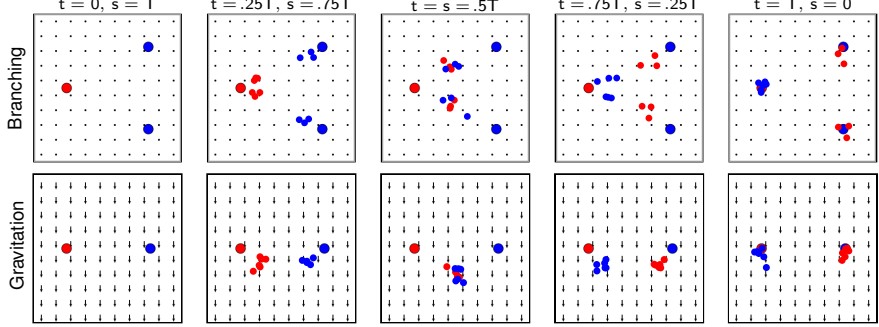

Figure 4: Trajectories in the point control problems. If SB exists in the acceleration space, particles will show a time-symmetric maneuver along with the initialization points (Blue & Red). Top: Our SSBM shows stochastic control that reaches multiple goals in a given time. Bottom: SSBM controls against gravity, successfully formulating SB regardless of apparent external forces.

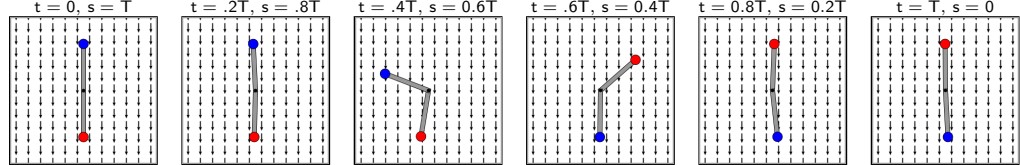

Figure 5: Demonstrations of forward control (red), and time-reversal control (blue) trajectories in the pendulum problem. The stochastic angular acceleration processes forms Schrödinger bridge.

In order to successfully model SB in the control problems, the maneuver between particles should be time-symmetric regardless of external drag and gravitational forces. The model-based control result in Fig. 4 shows that SSBM successfully induces Schrödinger bridge structure so that each position in the $t$ coordinates corresponds to positions of the reverse process in $T - t$ of the $s$ time coordinate. Notably, we observed that the induced SB structure was curved downwards when there is a gravitational force. This verifies that our SSBM is able to model path distributions with the principle of minimum control.

Table 4: Control performance measured by positional distances (5 runs, gravitation=1.0).

| goal dist. | Branching | Gravity | Pendulum | reverse | Branching | Gravity | Pendulum |
|---|---|---|---|---|---|---|---|
| DSB | 0.316±0.039 | **0.092±0.026** | 2.634±0.692 | DSB | 0.316±0.039 | 0.352±0.068 | **0.007±0.001** |
| SSBM | **0.255±0.023** | 0.151±0.024 | **0.230±0.079** | SSBM | **0.117±0.010** | **0.177±0.030** | 0.135±0.125 |

**A Pendulum.** Lastly, we considered a physical control problem of the pendulum environment. Since a pendulum is connected to a rod, this particular problem consists of variable gravitational force depending on the pendulum's angular position. In Fig. 5, the forward and reverse control agents swing the pendulum in a time-symmetrical manner, changing their angular positions throughout the time. Table 4 shows the numerical results compared to the DSB algorithm. In all cases, SSBM induced much more stable results in terms of forward and reverse control. By considering the success of the pendulum problem as reaching the top with the red pendulum by reaching the goal angles $\pi$ and $-\pi$, only SSBM was able to show the success of modeling SB in the task. Therefore, we conclude that our theoretical claims were also verified in the dynamic SB problems.

## 7 CONCLUSION

In this paper, we presented a symmetrization framework developed to solve both static and dynamic SBPs. Our approach allowed us to construct an optimal transport algorithm, with theoretical guarantees of linear convergence and monotonic improvements for a divergence. Based on the evidence, we claimed that the proposed SSBM method mitigate exaggerated displacement of couplings in Sinkhorn, by reflecting both sides of projection for each iteration. Compared to prior methods, our method empirically showed overall better stability in terms of learning and control.

The computational success of EOT methods was hinged upon the information geometrical properties of the KL divergence. Our work complements this key idea with a few more insights, implementing an efficient algorithm for finding the solution, which is more generally applicable to arbitrary space with Bregman projection (Bregman, 1967). A distinguishing feature—and concurrently a limitation—of SSBM is the dependency of the construction of target current drift on the conditional drift matching along with its corresponding samples. Such advancements in solving the IPF projections in non-compact high-dimensional spaces will help actualize the general application of SB methods in various subfields of machine learning.

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

Appendices for
# Symmetrized Schrödinger Bridge Matching

## ABBREVIATION AND NOTATION

| Abbreviation | Expansion |
|---|---|
| SB | Schrödinger Bridge |
| IPF | Iterative Proportional Fitting |
| PSIPF | Pseudo–Symmetric IPF |
| SDE | Stochastic Differential Equation |
| PDE | Partial Differential Equation |
| FBSDE | Forward-Bacward SDE |
| OT | Optimal Transport |
| FK | Feynman-Kac |
| HJE | Hamilton-Jacobi Equation |
| CDM | Conditional Drift Matching |
| KL | Kullback-Leibler |

| Notation | Usage |
|---|---|
| $\mu, \nu$ | marginal distributions |
| $\pi$ | a coupling of $\mu$ and $\nu$ |
| $n, \ell$ | IPF/PSIPF iteration |
| $\varphi, \psi$ | Schrödinger potentials |
| $\tilde{\varphi}, \tilde{\psi}$ | projections from a previous step |
| $\alpha, \beta$ | exponential of Schrödinger potentials |
| $\kappa$ | Contraction coefficient |
| $t, s$ | Forward/reversed time coordinates |
| $\rho(t, x)$ | probability distribution for $t$ |
| $H$ | (relative) entropy functions |
| $f, f^+, f^-$ | base / forward control / reverse control drifts |
| $v_\ell$ | current drift function |
| $g$ | diffusion matrix function |
| $(\Psi, \widehat{\Psi})$ | solution to SB PDEs |
| $(Y, \widehat{Y})$ | FK transformation for $(\Psi, \widehat{\Psi})$ |
| $\Phi$ | Time-symmetric |

## A    PROOFS

We assume $c_\lambda$ is bounded and two probability space $(\mathcal{X}, \mathcal{F}_1, \mu)$ and $(\mathcal{Y}, \mathcal{F}_2, \nu)$. where $\mathcal{F}_i$ denotes the $\sigma$-algebras. Then, suprema for Schrödinger potentials defined with $\|\varphi\|_\infty := \|\varphi\|_{L^\infty(\mathcal{X}, \mathcal{F}_1, \mu)}$ and $\|\psi\|_\infty := \|\varphi\|_{L^\infty(\mathcal{Y}, \mathcal{F}_2, \nu)}$. The proofs follows some results of (Sinkhorn, 1967; Sinkhorn & Knopp, 1967; Chen et al., 2016; Nutz & Wiesel, 2023; Carlier, 2022; Deng et al., 2023)

### A.1    PROPOSITION 3.1

First, we find boundedness of symmetrized Sinkhorn iterates.

**Lemma A.1.** *For every $\ell \geq 1$, the symmetrized Sinkhorn iterates are bounded such that*

$$\|\varphi_\ell\| \leq 2\|c_\lambda\|_\infty, \quad \|\psi_\ell\|_\infty \leq 2\|c_\lambda\|_\infty, \quad \|\tilde{\varphi}_\ell\| \leq 3\|c_\lambda\|_\infty, \quad \|\tilde{\psi}_\ell\|_\infty \leq 3\|c_\lambda\|_\infty \quad (17)$$

*and also*

$$\|\varphi_\ell \oplus \psi_\ell - c_\lambda\|_\infty \leq 5\|c_\lambda\|_\infty, \quad \|\tilde{\varphi}_{\ell+1} \oplus \psi_\ell - c_\lambda\|_\infty \leq 6\|c_\lambda\|_\infty, \quad \|\varphi_\ell \oplus \tilde{\psi}_{\ell+1} - c_\lambda\|_\infty \leq 6\|c_\lambda\|_\infty \quad (18)$$

*where $\|\cdot\|_\infty := \|\cdot\|_{L^\infty(\mathcal{X} \times \mathcal{Y}, \mathcal{F}_1 \otimes \mathcal{F}_2, \mu \otimes \nu)}$.*

*Proof.* For $(\mu \otimes \mu)$-a.e. $x_1, x_2 \in \mathcal{X}$ and $(\nu \otimes \nu)$-a.e. $y_1, y_2 \in \mathcal{Y}$, we can find

$$c_\lambda(x_1, y) \geq c_\lambda(x_2, y) - 2\|c_\lambda\|_\infty \ \forall y \in \mathcal{Y}, \qquad c_\lambda(x, y_1) \geq c_\lambda(x, y_2) - 2\|c_\lambda\|_\infty \ \forall x \in \mathcal{X}.$$

Using Eqs. (6) and (7), rewrite $(\varphi_\ell, \psi_\ell)$ as

$$\varphi_\ell = -\log \int_\mathcal{Y} e^{\psi_{\ell-1} - c_\lambda} d\nu - \log\sqrt{\kappa_\ell}, \qquad \psi_\ell = -\log \int_\mathcal{X} e^{\varphi_{\ell-1} - c_\lambda} d\mu - \log\sqrt{\kappa_\ell} \quad (19)$$

for $\ell = 1, \ldots, N-1$. Then, for $\ell = 1, \ldots, N-1$. Then, for $x_1, x_2 \in \mathcal{X}$, using the strong convexity of the exponential function,

$$\varphi_\ell(x_1) - \varphi_\ell(x_2) = \log \int_\mathcal{Y} e^{\psi_{\ell-1}(y) - c_\lambda(x_2, y)} \nu(dy) - \log \int_\mathcal{Y} e^{\psi_{\ell-1}(y) - c_\lambda(x_1, y)} \nu(dy)$$

$$\leq \log\left[\exp\{\sup_{y \in \mathcal{Y}}\left(c_\lambda(x_1, y) - c_\lambda(x_2, y)\right)\} \int_\mathcal{Y} e^{\psi_{\ell-1}(y) - c_\lambda(x_1, y)} \nu(dy)\right] \quad (20)$$

$$- \log \int_\mathcal{Y} e^{\psi_{\ell-1}(y) - c_\lambda(x_1, y)} \nu(dy) = \sup_{y \in \mathcal{Y}}[c_\lambda(x_1, y) - c_\lambda(x_2, y)] \leq 2\|c_\lambda\|_\infty.$$

Since $\mu(\varphi_\ell) = 0$, we have that $\sup_x \varphi(x) \geq 0$ and $\inf_x \varphi(x) \leq 0$. Therefore, this implies that $\|\varphi_\ell\|_\infty \leq 2\|c_\lambda\|_\infty$. We can follow the same procedure for $\psi_\ell$. These make $\|\tilde{\varphi}\|_\infty \leq 3\|c_\lambda\|_\infty$ and $\|\tilde{\psi}\|_\infty \leq 3\|c_\lambda\|_\infty$.

Similarly, we achieve

$$
\begin{aligned}
&\varphi_\ell(x_1) - \varphi_\ell(x_2) + \psi_\ell(y_1) - \psi_\ell(y_2) \\
&= \log \int_\mathcal{Y} e^{\psi_{\ell-1}(y) - c_\lambda(x_2,y)} \nu(\mathrm{d}y) - \log \int_\mathcal{Y} e^{\psi_{\ell-1}(y) - c_\lambda(x_1,y)} \nu(\mathrm{d}y) + \\
&\quad \log \int_\mathcal{X} e^{\varphi_{\ell-1}(x) - c_\lambda(x,y_2)} \mu(\mathrm{d}x) - \log \int_\mathcal{Y} e^{\varphi_{\ell-1}(x) - c_\lambda(x,y_1)} \mu(\mathrm{d}x) \\
&\leq \log\left[ \exp\{ \sup_{y\in\mathcal{Y}} (c_\lambda(x_1,y) - c_\lambda(x_2,y))\} \int_\mathcal{Y} e^{\psi_{\ell-1}(y) - c_\lambda(x_1,y)} \nu(\mathrm{d}y) \right] - \log \int_\mathcal{Y} e^{\psi_{\ell-1}(y) - c_\lambda(x_1,y)} \nu(\mathrm{d}y) + \\
&\quad \log\left[ \exp\{ \sup_{x\in\mathcal{X}} (c_\lambda(x,y_1) - c_\lambda(x,y_2))\} \int_\mathcal{Y} e^{\varphi_{\ell-1}(y) - c_\lambda(x,y_1)} \mu(\mathrm{d}x) \right] - \log \int_\mathcal{X} e^{\varphi_{\ell-1}(x) - c_\lambda(x,y_1)} \mu(\mathrm{d}x) \\
&= \sup_{y\in\mathcal{Y}} [c_\lambda(x_1,y) - c_\lambda(x_2,y)] + \sup_{x\in\mathcal{X}} [c_\lambda(x,y_1) - c_\lambda(x,y_2)] \leq 4\|c_\lambda\|_\infty.
\end{aligned}
\tag{21}
$$

By the Radon-Nikodym theorem, the derivative satisfy $\iint_{\mathcal{X}\times\mathcal{Y}} e^{\varphi_\ell \oplus \psi_\ell - c_\lambda} \mathrm{d}(\mu\otimes\nu) = 0$, and this implies that $\forall y \; \sup_x \varphi_\ell(x) + \psi_\ell(y) - c_\lambda(x,y) \geq 0$ and $\forall y \; \inf_x \varphi_\ell(x) + \psi_\ell(y) - c_\lambda(x,y) \leq 0$ (this holds same for the $y$ case). Therefore, we have proved that $\|\varphi_\ell \oplus \psi_\ell - c_\lambda\|_\infty \leq 5\|c_\lambda\|_\infty$. Furthermore, the definition of $(\tilde{\varphi}_\ell, \tilde{\psi}_\ell)$ yields $\|\tilde{\varphi}_{\ell+1} \oplus \psi_\ell - c_\lambda\|_\infty \leq \|\varphi_\ell \oplus \psi_\ell - c_\lambda\|_\infty + \|c\|_\infty \leq 6\|c_\lambda\|_\infty$ and also $\|\varphi_\ell \oplus \tilde{\psi}_{\ell+1} - c_\lambda\|_\infty \leq 6\|c_\lambda\|_\infty$, thereby completing the proof. $\quad\square$

Next, we introduce the general inequality for the dual functional $F$.

**Lemma A.2.** *Define:*

$$
\partial_\varphi F(\varphi,\psi) := 1 - \int_\mathcal{Y} e^{\varphi\oplus\psi - c_\lambda} \mathrm{d}\nu, \qquad \partial_\psi F(\varphi,\psi) := 1 - \int_\mathcal{X} e^{\varphi\oplus\psi - c_\lambda} \mathrm{d}\mu.
\tag{22}
$$

*If both $\varphi \oplus \psi - c_\lambda \geq -a$ and $\varphi' \oplus \psi' - c_\lambda \geq -a$ for some $a \in \mathbb{R}$, we have*

$$
\begin{aligned}
F(\varphi',\psi') - F(\varphi,\psi) \geq{}& \int_\mathcal{X} \partial_\varphi F(\varphi',\psi')[\varphi'-\varphi]\mathrm{d}\mu + \int_\mathcal{Y} \partial_\psi F(\varphi',\psi')[\psi'-\psi]\mathrm{d}\nu \\
&+ \frac{e^{-a}}{2} \|(\varphi-\varphi')\oplus(\psi-\psi')\|_{L^2(\mu\otimes\nu)}
\end{aligned}
\tag{23}
$$

*where $\|\cdot\|_{L^2(\mu\otimes\nu)}$ is a shorthand notation for $\|\cdot\|_{L^2(\mathcal{X}\times\mathcal{Y}, F_1\otimes F_2, \mu\otimes\nu)}$.*

*Proof.* Recall the strong convexity of the exponential function for $x \in [-a,\infty]$ and some constant $a$,

$$
e^x - e^y \geq (x-y)e^y + \frac{e^{-y}}{2}|x-y|^2, \qquad x,y \in [-a,\infty].
$$

Then, we have

$$
\begin{aligned}
&F(\varphi',\psi') - F(\varphi,\psi) \\
&= \mu(\varphi'-\varphi) + \nu(\psi'-\psi) + \iint_{\mathcal{X}\times\mathcal{Y}} \left( e^{\varphi\oplus\psi - c_\lambda} - e^{\varphi'\oplus\psi' - c_\lambda} \right) \mathrm{d}(\mu\otimes\nu) \\
&\geq \mu(\varphi'-\varphi) + \nu(\psi'-\psi) + \iint_{\mathcal{X}\times\mathcal{Y}} (\varphi\oplus\psi - \varphi'\oplus\psi')e^{\varphi'\oplus\psi' - c_\lambda} \mathrm{d}(\mu\otimes\nu) \\
&\quad + \frac{e^a}{2} \iint_{\mathcal{X}\times\mathcal{Y}} \|\varphi\otimes\psi - \varphi'\otimes\psi'\|_2^2 \mathrm{d}(\mu\otimes\nu) \\
&= \int_\mathcal{X} \partial_\varphi F(\varphi',\psi')[\varphi'-\varphi]\mathrm{d}\mu + \int_\mathcal{Y} \partial_\psi F(\varphi',\psi')[\psi'-\psi]\mathrm{d}\nu + \frac{e^{-a}}{2}\|(\varphi-\varphi')\oplus(\psi-\psi')\|_{L^2(\mu\otimes\nu)}.
\end{aligned}
\tag{24}
$$

The proof is complete. $\quad\square$

To advance the argument, we establish that change the dual functional $F$ with respect to concurrent projection are subject to a lower bound of $\|\tilde{\varphi}_{\ell+1} - \psi_\ell\|_{L^2(\mu)} + \|\tilde{\psi}_{\ell+1} - \psi_\ell\|_{L^2(\nu)}$, inspired by Lemma 3.1 of (Carlier, 2022). Specifically, we use the inequality such that $\forall (a, b) \in [-M, M]^2$

$$e^b - e^a - e^a(b - a) \geq \frac{e^{-M}}{2}(b - a)^2, \qquad |e^b - e^a| \leq e^M |b - a| \tag{25}$$

**Lemma A.3.** *For $\ell \in \mathbb{N}$, the iterates satisfy the following inequality:*

$$\frac{1}{2}\big(F(\tilde{\varphi}_{\ell+1}, \psi_\ell) + F(\varphi_\ell, \tilde{\psi}_{\ell+1})\big) - F(\varphi_\ell, \psi_\ell) \geq \frac{\sigma}{2}\big(\|\tilde{\varphi}_{\ell+1} - \varphi_\ell\|_{L^2(\mu)}^2 + \|\tilde{\psi}_{\ell+1} - \psi_\ell\|_{L^2(\nu)}^2\big) \tag{26}$$

*where $\sigma = e^{-5\|c_\lambda\|_\infty}$.*

*Proof.* From the fact from Sinkhorn iterates that $\tilde{\varphi}_{\ell+1} - \varphi_\ell$ and $\tilde{\psi}_{\ell+1} - \varphi_\ell$ has zero mean against $\mu$ and $\nu$, we get

$$
\begin{aligned}
F(\tilde{\varphi}_{\ell+1}, \psi_\ell) - F(\varphi_\ell, \psi_\ell) &= \iint_{\mathcal{X} \times \mathcal{Y}} (e^{\tilde{\varphi}_{\ell+1}} - e^{\varphi_\ell}) e^{-c_\lambda} e^{\psi_\ell} \mathrm{d}(\mu \otimes \nu) \\
&\geq \iint_{\mathcal{X} \times \mathcal{Y}} (\varphi_\ell - \tilde{\varphi}_{\ell+1}) e^{\tilde{\varphi}_{\ell+1}} e^{-c_\lambda} e^{\psi_\ell} \mathrm{d}(\mu \otimes \nu) \\
&\quad + \frac{e^{-2\|c_\lambda\|_\infty}}{2} \int_{\mathcal{X}} ((\varphi_\ell - \tilde{\varphi}_{\ell+1}))^2 e^{-c_\lambda} e^{\psi_\ell} \mathrm{d}(\mu \otimes \nu) \\
&\geq \text{constant} \cdot \int_{\mathcal{X}} (\varphi_\ell - \tilde{\varphi}_\ell) \mathrm{d}\mu + \frac{e^{-5\|c_\lambda\|_\infty}}{2} \iint_{\mathcal{X}} (\tilde{\varphi}_{\ell+1} - \varphi_\ell)^2 \mathrm{d}\mu \\
&= \frac{e^{-5\|c_\lambda\|_\infty}}{2} \iint_{\mathcal{X}} (\tilde{\varphi}_{\ell+1} - \varphi_\ell)^2 \mathrm{d}\mu
\end{aligned}
\tag{27}
$$

And similarly,

$$F(\varphi_\ell, \tilde{\psi}_{\ell+1}) - F(\varphi_\ell, \psi_\ell) \geq \frac{e^{-5\|c_\lambda\|_\infty}}{2}, \tag{28}$$

thereby completing the proof. $\qquad \square$

Since the dual function $F$ is strictly concave, arbitrary Sinkhorn iterates *e.g.* $\psi_{2n+1} = \mathrm{argmax}_\psi F(\varphi_{2n}, \cdot)$ is not only the maximizing argument for $\psi_{2n-1}$, but also the general inequality

$$F(\varphi, \psi_{2n-1}) \geq F(\varphi, \psi_{2n+1}) \ \forall n \in \mathbb{N} \tag{29}$$

holds for arbitrary $\varphi \in L^1(\mu)$, under the condition the integration $\iint_{\mathcal{X} \times \mathcal{Y}} \varphi \oplus \psi_{2n+1} \mathrm{d}(\mu \otimes \nu)$ equals to that of $\varphi_{2n}$. In our case, this makes the following inequality:

$$F(\tilde{\varphi}_{\ell+1}, \psi_\ell) \leq F(\tilde{\varphi}_{\ell+1}, \tilde{\psi}_{\ell+1} - \log \kappa_{\ell+1}) = F(\varphi_{\ell+1}, \psi_{\ell+1}). \tag{30}$$

Hence, we arrive at

$$\frac{1}{2}\big(F(\tilde{\varphi}_{\ell+1}, \psi_\ell) + F(\varphi_\ell, \tilde{\psi}_{\ell+1})\big) \leq F(\varphi_{\ell+1}, \psi_{\ell+1}). \tag{31}$$

Finally, we prove Proposition 3.1 by the following lemma.

**Lemma A.4.** *Suppose the existence of optimality $(\varphi_*, \psi_*)$ specified with $\mu(\varphi_*) = \nu(\psi_*) = 0$. Then, the iterates of the symmetrized Sinkhorn satisfy*

$$F(\varphi_*, \psi_*) - F(\varphi_\ell, \psi_\ell) \leq k^\ell \big(F(\varphi_*, \psi_*) - F(\varphi_0, \psi_0)\big) \tag{32}$$

*where $k := 1 - e^{-22\|c_\lambda\|_\infty}$.*

*Proof.* Using Lemma A.2 for $a = 5\|c_\lambda\|_\infty$, we have

$$
\begin{aligned}
F(\varphi_\ell, \psi_\ell) - F(\varphi_*, \psi_*) &\geq \int_{\mathcal{X}} \partial_\varphi F(\varphi_\ell, \psi_\ell)[\varphi_\ell - \varphi_*]\mathrm{d}\mu + \int_{\mathcal{Y}} \partial_\psi F(\varphi_\ell, \psi_\ell)[\psi_\ell - \psi_*]\mathrm{d}\nu \\
&\quad + \frac{\exp(-5\|c_\lambda\|_\infty)}{2} \Big\|(\varphi_* - \varphi_\ell) \oplus (\psi_* - \psi_\ell)\Big\|_{L^2(\mu \otimes \nu)}.
\end{aligned}
\tag{33}
$$

Since $\partial_\varphi F(\tilde{\varphi}_{\ell+1}, \psi_\ell)$ is a deterministic scalar and $\mu(\varphi_\ell) = \mu(\varphi_*) = 0$, we can say

$$\int_{\mathcal{X}} \partial_\varphi F(\tilde{\varphi}_{\ell+1}, \psi_\ell)[\varphi_\ell - \varphi_*]\mathrm{d}\mu = 0 \tag{34}$$

Therefore, for the first integral of Eq. (33) is decomposed as

$$\begin{aligned}
&\int_{\mathcal{X}} \partial_\varphi F(\varphi_\ell, \psi_\ell)[\varphi_\ell - \varphi_*]\mathrm{d}\mu \\
&= \int_{\mathcal{X}} [\partial_\varphi F(\varphi_\ell, \psi_\ell) - \partial_\varphi F(\tilde{\varphi}_{\ell+1}, \psi_\ell)][\varphi_\ell - \varphi_*]\mathrm{d}\mu \\
&\geq -\frac{1}{2\sigma_1}\|\partial_\varphi F(\varphi_\ell, \psi_\ell) - \partial_\varphi F(\tilde{\varphi}_{\ell+1}, \psi_\ell)\|^2_{L^2(\mu)} - \frac{\sigma_1}{2}\|\varphi_\ell - \varphi_*\|^2_{L^2(\mu)}
\end{aligned} \tag{35}$$

where $\sigma_1 := e^{-5\|c_\lambda\|_\infty}$ the inequality follows from Hölder's inequality and Young's inequality. By the same process for the second integral of Eq. (33), using $\partial_\psi F(\varphi_\ell, \tilde{\psi}_{\ell+1})$, we achieve

$$\begin{aligned}
F(\varphi_*, \psi_*) - F(\varphi_\ell, \psi_\ell) \leq &\frac{1}{2\sigma_1}\|\partial_\varphi F(\varphi_\ell, \psi_\ell) - \partial_\varphi F(\tilde{\varphi}_{\ell+1}, \psi_\ell)\|^2_{L^2(\mu)} \\
&+ \frac{1}{2\sigma_1}\|\partial_\psi F(\varphi_\ell, \psi_\ell) - \partial_\psi F(\varphi_\ell, \tilde{\psi}_{\ell+1})\|^2_{L^2(\mu)}
\end{aligned} \tag{36}$$

Note that for $x \in \mathcal{X}$,

$$\begin{aligned}
|\partial_\varphi F(\varphi_\ell, \psi_\ell)(x) - \partial_\psi F(\tilde{\varphi}_{\ell+1}, \psi_\ell)(x)| &\leq \int_{\mathcal{Y}} |e^{\tilde{\varphi}_{\ell+1} \oplus \psi_\ell - c_\lambda} - e^{\varphi_\ell \oplus \psi_\ell - c_\lambda}|\mathrm{d}\nu \\
&\leq e^{6\|c_\lambda\|_\infty} \int_{\mathcal{Y}} |\tilde{\varphi}_{\ell+1} \oplus \psi_\ell - \varphi_\ell \oplus \psi_\ell|\mathrm{d}\nu \\
&= \frac{1}{\sigma_2}|\tilde{\varphi}_{\ell+1}(x) - \varphi_\ell(x)|.
\end{aligned} \tag{37}$$

where the second inequality follows by the boundedness and Lipchitz continuity of the exponential function in certain region such that $e^a - e^b \leq e^M|b - a|$ for $a, b \leq M$; $\sigma_2 := e^{-6\|c_\lambda\|_\infty}$.

Combining Eqs. (36) and (37) after finding the inequality for $|\partial_\varphi F(\varphi_\ell, \psi_\ell)(x) - \partial_\psi F(\varphi_\ell, \tilde{\psi}_{\ell+1})(x)|$ with the same manner, we conclude that

$$\begin{aligned}
F(\varphi_*, \psi_*) - F(\varphi_\ell, \psi_\ell) &\leq \frac{1}{2\sigma_1\sigma_2^2}\Big(\|\tilde{\varphi}_{\ell+1} - \varphi_\ell\|^2_{L^2(\mu)} + \|\tilde{\psi}_{\ell+1} - \psi_\ell\|^2_{L^2(\nu)}\Big) \\
&\leq \frac{1}{2\sigma_1\sigma_2^2}\Big(\|\varphi_{\ell+1} - \varphi_\ell\|^2_{L^2(\mu)} + \|\psi_{\ell+1} - \psi_\ell\|^2_{L^2(\nu)}\Big) \\
&\leq \frac{1}{\sigma_1^2\sigma_2^2}\Big(\frac{1}{2}\big(F(\tilde{\varphi}_{\ell+1}, \psi_\ell) + F(\varphi_\ell, \tilde{\psi}_{\ell+1})\big) - F(\varphi_\ell, \psi_\ell)\Big) \\
&\leq \frac{1}{\sigma_1^2\sigma_2^2}\Big(F(\varphi_{\ell+1}, \psi_{\ell+1}) - F(\varphi_\ell, \psi_\ell)\Big)
\end{aligned} \tag{38}$$

Denoting the suboptimality gap $\Delta_\ell = F(\varphi_*, \psi_*) - F(\varphi_\ell, \psi_\ell)$ and $k_\varepsilon := 1 - e^{-22\|c_\lambda\|_\infty} \in (0, 1)$, we have

$$\Delta_\ell \leq \frac{1}{\sigma_1^2\sigma_2^2}(\Delta_\ell - \Delta_{\ell+1}). \tag{39}$$

In other words, we apparently reach the contraction property as follows

$$\Delta_{\ell+1} \leq (1 - \sigma_1^2\sigma_2^2)\Delta_\ell \leq (1 - \sigma_1^2\sigma_1^2)^{\ell+1}\Delta_0 \tag{40}$$

Denote $k_\varepsilon := 1 - e^{-22\|c_\lambda\|_\infty} \in (0, 1)$. We hereby complete the proposition for any $\ell \geq 1$

$$\Delta_\ell \leq k^\ell \Delta_0. \tag{41}$$

This proves the lemma and Proposition 3.1. $\qquad\square$

## A.2 PROPOSITION 3.2

We mainly relate this result to (Nutz, 2021). By the construction, we have

$$\frac{\mathrm{d}\mu_\ell}{\mathrm{d}\mu}(x) = \int_{\mathcal{Y}} \frac{\mathrm{d}\pi_\ell}{\mathrm{d}\mu \otimes \nu}(x,y)\nu(\mathrm{d}y) = \int_{\mathcal{Y}} e^{\varphi_\ell(x)+\psi_\ell(y)-c_\lambda(x,y)}\nu(\mathrm{d}y)$$
$$= e^{\varphi_\ell(x)} \int_{\mathcal{Y}} e^{\psi_\ell(y)-c_\lambda(x,y)}\nu(\mathrm{d}y) = e^{\varphi_\ell(x)-\tilde{\varphi}_{\ell+1}(x)} \tag{42}$$

where we used the definition $\tilde{\varphi}_{\ell+1}$ in the last step. As a result, we have

$$\frac{\mathrm{d}\mu_\ell}{\mathrm{d}\mu} = e^{\varphi_\ell-\tilde{\varphi}_{\ell+1}}, \qquad \frac{\mathrm{d}\nu_\ell}{\mathrm{d}\nu} = e^{\psi_\ell-\tilde{\psi}_{\ell+1}}. \tag{43}$$

And the relative entropy is represented with

$$H(\mu|\mu_\ell) = \mu(\varphi_\ell - \tilde{\varphi}_\ell), \quad H(\nu|\nu_\ell) = \nu(\psi_\ell - \tilde{\psi}_\ell). \tag{44}$$

Since the additive operation in the symmetrized Sinkhorn does not change the essential property of projection, the well-known monotony property (Lemma 6.7 of Nutz, 2021):

$$H(\mu|\mu_\ell) \leq H(\mu|\mu_{\ell+2}), \qquad H(\nu|\nu_\ell) \leq H(\nu|\nu_{\ell+2}) \tag{45}$$

holds, and thus both sequences:

$$\{H(\mu|\mu_{2t}) + H(\mu|\mu_{2t}) + H(\nu|\nu_{2t+1}) + H(\mu|\nu_{2t+1})\}_{t\geq 0},$$
$$\{H(\mu|\mu_{2t-1}) + H(\mu|\mu_{2t-1}) + H(\nu|\nu_{2t}) + H(\mu|\nu_{2t})\}_{t\geq 0},$$

are monotone decreasing. Using the definition of $\pi_\ell$ the relative entropy can be drawn:

$$\begin{aligned}
H(\pi_*|\pi_{\ell+2}) &- H(\pi_*|\pi_\ell) \\
&= [H(\pi_*|\pi_{\ell+2}) - H(\pi_*|\pi_{\ell+1})] - [H(\pi_*|\pi_{\ell+1}) - H(\pi_*|\pi_\ell)] \\
&= \mu(\varphi_{\ell+1} - \varphi_{\ell+2}) + \nu(\psi_{\ell+1} - \psi_{\ell+2}) + \mu(\varphi_\ell - \varphi_{\ell+1}) + \nu(\psi_\ell - \psi_{\ell+1}) \\
&= \mu(\varphi_{\ell+1} - \tilde{\varphi}_{\ell+2}) + \nu(\psi_{\ell+1} - \tilde{\psi}_{\ell+2}) + \mu(\varphi_\ell - \tilde{\varphi}_{\ell+1}) + \nu(\psi_\ell - \tilde{\psi}_{\ell+1}) - \log \kappa_{\ell+1}\kappa_{\ell+2} \\
&= -[H(\mu|\mu_{\ell+1}) + H(\nu|\nu_{\ell+1}) + H(\mu|\mu_\ell) + H(\nu|\nu_\ell) + \log \kappa_{\ell+1}\kappa_{\ell+2}].
\end{aligned} \tag{46}$$

The Birkhoff-Bushell theorem (Birkhoff, 1957; Bushell, 1973) predicts measure contraction property such that $\log \kappa_\ell \leq 0$. Since the suboptimality gap gradually gets minimal by Proposition 3.1, $\log \kappa_\ell$ is monotone increasing to 0. Given that $\pi_\ell$ converges to $\pi_*$, and that monotony of Eq. (46), there must exist some $\ell' \in \mathbb{N}$ such that

$$H(\mu|\mu_\ell) + H(\nu|\nu_\ell) \geq -\log \kappa_\ell, \qquad \ell \geq \ell'. \tag{47}$$

Therefore, for sufficiently large $\ell \geq \ell' - 1$

$$H(\pi_*|\pi_{\ell+1}) - H(\pi_*|\pi_\ell) = -[H(\mu|\mu_\ell) + H(\nu|\nu_\ell) + \log \kappa_\ell] \geq 0 \tag{48}$$

holds, thereby completing the proof.

## A.3 PROPOSITION 4.1

Using the fact that each SDE and time reversal drifts respectively follow the forward and backward Kolmogorov (or Fokker–Planck) equations, we write

$$\frac{\partial \rho}{\partial t} = -\nabla \cdot (f^+ \rho) + \nabla^2 \cdot \left(\tfrac{1}{2}gg^\mathsf{T}\rho\right), \tag{49a}$$

$$\frac{\partial \rho}{\partial t} = \nabla \cdot (f^- \rho) - \nabla^2 \cdot \left(\tfrac{1}{2}gg^\mathsf{T}\rho\right), \tag{49b}$$

Adding two equations, we achieve the continuity equation:

$$\frac{\partial \rho}{\partial t} + \nabla \cdot (v\rho) = 0. \tag{50}$$

where $v = \frac{1}{2}f^+ - f^-$. Subtracting the forward and backward Kolmogorov equations also derives another fundamental identity

$$(f^+ + f^-)\rho = \nabla \cdot (gg^{\mathsf{T}}\rho). \tag{51}$$

Moreover, we can derive an explicit form of the score function:

$$(f^+ + f^-)\rho = \nabla \cdot (gg^{\mathsf{T}}\rho) \quad \Leftrightarrow \quad f^+ + f^- = gg^{\mathsf{T}}\nabla \log \rho + \nabla \cdot gg^{\mathsf{T}}$$
$$\Leftrightarrow \quad \nabla \log \rho = (gg^{\mathsf{T}})^{-1}(f^+ + f^- - \nabla \cdot gg^{\mathsf{T}}),$$

which can be considered as a general derivation of the Nelson's first equation of the stochastic calculus (Nelson, 2001) for multivariate SDEs. Generalizing the Nelson's duality, define the *current* and *osmotic* drifts for measures $\mathbb{P}$ and $\mathbb{Q}$:

$$v^{\mathbb{P}} = \frac{f_{\mathbb{P}}^+ - f_{\mathbb{P}}^-}{2}, \quad u^{\mathbb{P}} = \frac{f_{\mathbb{P}}^+ + f_{\mathbb{P}}^-}{2}, \quad v^{\mathbb{Q}} = \frac{f_{\mathbb{Q}}^+ - f_{\mathbb{Q}}^-}{2}, \quad u^{\mathbb{Q}} = \frac{f_{\mathbb{Q}}^+ + f_{\mathbb{Q}}^-}{2}, \tag{52}$$

where $(f_{\mathbb{P}}^+, f_{\mathbb{P}}^-)$ is drawn from SB-FBSDE (9) and $(f_{\mathbb{P}}^-, f_{\mathbb{Q}}^-)$ is the base drifts, assuming that the .

Under they Girsanov theorem (Øksendal, 2003), we observe that

$$H(\mathbb{P}|\mathbb{Q}) = H(\mathbb{P}_0|\mathbb{Q}_0) + \mathbb{E}_{\mathbb{P}}\left[\int_0^T \frac{1}{2}\|f_{\mathbb{P}}^+ - f_{\mathbb{Q}}^+\|^2 \mathrm{d}t\right]$$
$$= H(\mathbb{P}_T|\mathbb{Q}_T) + \mathbb{E}_{\mathbb{P}}\left[\int_0^T \frac{1}{2}\|f_{\mathbb{P}}^- - f_{\mathbb{Q}}^-\|^2 \mathrm{d}t\right]$$

Combining above two equations, we get

$$H(\mathbb{P}|\mathbb{Q}) = \frac{1}{2}H(\mathbb{P}_0|\mathbb{Q}_0) + \frac{1}{2}H(\mathbb{P}_T|\mathbb{Q}_T) + \mathbb{E}_{\mathbb{P}}\left[\int_0^T \frac{1}{4}\|f_{\mathbb{P}}^+ - f_{\mathbb{Q}}^+\|^2 + \frac{1}{4}\|f_{\mathbb{P}}^- - f_{\mathbb{Q}}^-\|^2 \mathrm{d}t\right]$$
$$= \frac{1}{2}H(\mathbb{P}_0|\mathbb{Q}_0) + \frac{1}{2}H(\mathbb{P}_T|\mathbb{Q}_T) + \mathbb{E}_{\mathbb{P}}\left[\int_0^T \frac{1}{4}\|v_t^{\mathbb{P}} - v_t^{\mathbb{Q}}\|^2 + \frac{1}{4}\|u_t^{\mathbb{P}} - u_t^{\mathbb{Q}}\|^2 \mathrm{d}t\right].$$

Knowing that $\mathbb{P}_0 = \mathbb{Q}_0 = \mu$ and $\mathbb{P}_T = \mathbb{Q}_T = \nu$, and plugging SB-FBSDE derived in Appendix B, we achieve

$$H(\mathbb{P}|\mathbb{Q}) = \mathbb{E}_{\mathbb{P}}\left[\int_0^T \frac{1}{8}\|\nabla \log \Psi - \nabla \log \widehat{\Psi}\|_{gg^{\mathsf{T}}}^2 + \frac{1}{8}\|\nabla \log \Psi + \nabla \log \widehat{\Psi}\|_{gg^{\mathsf{T}}}^2 \mathrm{d}t\right] \tag{53}$$

This is equivalent to

$$\int_0^T \int_{\mathbb{R}^d}\left(\frac{1}{2}\|(v_t - f_t)(x)\|^2 + \frac{1}{8}\|\nabla \log \rho_t(x)\|_{gg^{\mathsf{T}}}^2\right)\rho_t(x)\mathrm{d}x\mathrm{d}t.$$

We note that when $(\Psi, \widehat{\Psi})$ satisfy Eq. (10), and also with the reversed time coordinate $s$, the continuity equation (50) in a logarithmic form.

Finally, we get the constrained problem:

$$\inf_{(\tilde{\rho}, \tilde{v})} \int_0^T \int_{\mathbb{R}^d}\left(\frac{1}{2}\|\tilde{v}(t, x) - f(t, x)\|^2 + \frac{1}{8}\|\nabla \log \tilde{\rho}(t, x)\|_{gg^{\mathsf{T}}}^2\right)\tilde{\rho}(x, t)\, \mathrm{d}t\, \mathrm{d}x,$$
$$\text{such that} \quad \frac{\partial \tilde{\rho}}{\partial t} + \nabla \cdot (\tilde{v}\tilde{\rho}) = 0, \ \tilde{\rho}(0, \cdot) \equiv \mu, \ \tilde{\rho}(T, \cdot) \equiv \nu.$$

To solve this problem, we convert the problem to the Lagrangian function:

$$\mathcal{L}(\rho, v) = \int_0^T \int_{\mathbb{R}^d}\frac{1}{2}\|\tilde{v}(t, x) - v(t, x)\|^2\tilde{\rho}(t, x) + \frac{1}{8}\|\nabla \log \tilde{\rho}(t, x)\|_{gg^{\mathsf{T}}}^2\tilde{\rho}(t, x)$$
$$+ \lambda(t, x)\left(\frac{\partial \tilde{\rho}}{\partial t} + \nabla \cdot (\tilde{v}\tilde{\rho})\right)\mathrm{d}x\, \mathrm{d}t$$

where $\lambda$ is $C^{1,2}$-Lagrangian multiplier. After integration by part, assuming that limits for $x \to \infty$ are zero, and observing that the boundary values are constant over $\Pi(\mu, \nu)$, we resort to the following problem:

$$\inf_{(\tilde{\rho}, \tilde{v}) \in \mathcal{P} \times \mathcal{V}} \int_{\mathbb{R}^n} \int_0^T\left[\frac{1}{2}\|v(t, x) - \tilde{v}(t, x)\|^2 + \frac{1}{8}\|\nabla \log \tilde{\rho}(t, x)\|^2 + \left(-\frac{\partial \lambda}{\partial t} - \nabla \lambda \cdot \tilde{v}\right)\right]\tilde{\rho}(t, x)\mathrm{d}t\mathrm{d}x. \tag{54}$$

Pointwise minimization with respect to $\tilde{v}$ for each fixed flow of probability densities $\tilde{\rho}$ gives

$$v_\rho^*(x,t) = v(x,t) + \nabla\lambda(x,t).$$

Plugging this form of the optimal control into Eq. (54), we get the functional of $\tilde{\rho} \in \mathcal{P}$:

$$\mathcal{J}(\tilde{\rho}) = -\int_{\mathbb{R}^n} \int_0^T \left[\frac{\partial\lambda}{\partial t} + v \cdot \nabla\lambda + \frac{1}{2}\|\nabla\lambda\|^2 + \frac{1}{8}\|\nabla\log\tilde{\rho}(t,x)\|_{gg^\mathsf{T}}^2\right]\tilde{\rho}(t,x) + \mathrm{d}t\mathrm{d}x$$

Utilizing the existence and uniqueness of SDE solutions, and by using the relation found in Eqs. (52) and (53), we get

$$\frac{\partial\Phi(t,x)}{\partial t} + v_t \cdot \nabla\Phi(t,x) = \frac{1}{4}\left\|\nabla\log\Psi(t,x)\right\|_{gg^\mathsf{T}}^2 + \frac{1}{4}\left\|\nabla\log\widehat{\Psi}(s,\bar{x})\right\|_{gg^\mathsf{T}}^2$$

$$\Phi(t,x) = \nicefrac{1}{2}\{\log\Psi(t,x) - \log\widehat{\Psi}(s,x)\}, \quad s := T - t$$

which completes the proof.

# B    DERIVATIONS OF THE SB-FBSDE

In this section, we introduce SB-FBSDE, a control dynamic formulation for describing SB (Chen et al., 2022; Liu et al., 2022a). The formulation allows us to link the static SB problem into real-world physical problems. Notably, we present detailed derivations on SB-FBSDE in multi-variate case.

## B.1    PRELIMINARIES FOR SB-FBSDE

First, we present the Itô's lemma.

**Lemma B.1** (Itô's lemma (Itô, 1951)). *Let $X_t$ be the solution to the Itô SDE:*

$$\mathrm{d}X_t = f(t, X_t)\,\mathrm{d}t + g(t, X_t)\mathrm{d}W_t$$

*Then, the stochastic process $u(t, X_t)$, where $u \in C^{1,2}([0,T],\mathbb{R}^d)$, is also an Itô process satisfying*

$$\mathrm{d}u(t, X_t) = \frac{\partial u(t, X_t)}{\partial t}\mathrm{d}t + \left[\nabla u(t, X_t)^\mathsf{T} f(t, X_t) + \frac{1}{2}\operatorname{Tr}[gg^\mathsf{T}(t, X_t)\nabla^2 u(t, X_t)]\mathrm{d}W_t.\right]\mathrm{d}t \tag{55}$$
$$+ [\nabla u(t, X_t)^\mathsf{T} g(t, X_t)]\mathrm{d}W_t$$

Next, we introduce Feynman-Kac Lemma, which predicts potentials, or value functions

**Lemma B.2** (Nonlinear Feynman-Kac (Exarchos & Theodorou, 2018; Yong & Zhou, 1999)). *Let $u \equiv u(x,t)$ be a function that is twice continuously differentiable in $x \in \mathbb{R}^d$ and once differentiable in $t \in [0,T]$, i.e., $u \in C^{1,2}([0,T],\mathbb{R}^d)$. Consider the following second-order parabolic PDE,*

$$\frac{\partial u}{\partial t} + \frac{1}{2}\operatorname{Tr}(gg^\mathsf{T}\nabla^2 u) + \nabla u^\mathsf{T} f(t,x) + h(t,x,u,g^\mathsf{T}\nabla u) = 0, u(T,\cdot) \equiv \tau(\cdot), \tag{56}$$

*where the functions $f$, $g$, $h$, and $\tau$ satisfy proper regularity conditions. Specifically, ① $f$, $g$, $h$, and $\tau$ are continuous, ② $f(t,x)$ and $g(t,x)$ are uniformly Lipschitz in $x$, and ③ $h(t,x,y,z)$ satisfies quadratic growth condition in $z$. Then, Eq. (56) exists a unique solution $v = u$ such that the following stochastic representation (known as the nonlinear Feynman-Kac transformation) holds:*

$$Y_t = u(t, X_t), \qquad Z_t = g^\mathsf{T}(t, X_t)\nabla u(t, X_t) \tag{57}$$

*where $(X_t, Y_t, Z_t)$ are the unique adapted solutions to the following FBSDEs:*

$$\mathrm{d}X_t = f(X_t, t)\mathrm{d}t + g(X_t, t)\mathrm{d}W_t, \qquad X_0 = x_0, \tag{58a}$$
$$\mathrm{d}Y_t = -h(X_t, Y_t, Z_t, t)\mathrm{d}t + Z_t^\mathsf{T}\mathrm{d}W_t, \qquad Y_T = \tau(X_T). \tag{58b}$$

*The original deterministic PDE solution $v(x,t)$ can be recovered by taking conditional expectations:*

$$\mathbb{E}[Y_t|X_t = x] = u(t,x), \qquad \mathbb{E}[Z_t|X_t = x] = g(t,x)^\mathsf{T}\nabla u(t,x).$$

Lemma B.2 establishes an intriguing connection between a certain class of (nonlinear) PDEs (56) and FBSDEs (58) via the nonlinear FK transformation Eq. (57).

**SB-FBSDE.** SB-FBSDE is a class of probabilistic models that, inspired by optimal control and neural differential equations (Chen et al., 2022; Kirk, 1970; Chen et al., 2018), adopts Lemma B.2 to generalize the score-based diffusion models. Since the PDEs $(\partial_t \Psi, \partial_t \widehat{\Psi})$ in the SB Eq. (10) are both of the parabolic form Eq. (56), one can apply Lemma B.2 and derive the corresponding FBSDEs. In this appendix, we present a more general expression of SB-FBSDE than previous literature:

$$
\begin{cases}
\mathrm{d}X_t = f_t^+ \mathrm{d}t + g^{\mathsf{T}}\mathrm{d}W_t = (f_t + gZ_t)\mathrm{d}t + g^{\mathsf{T}}\mathrm{d}W_t & \text{(59a)} \\[2mm]
\mathrm{d}Y_t = \frac{1}{2}\|Z_t\|^2 \mathrm{d}t + Z_t^{\mathsf{T}}\, \mathrm{d}W_t & \text{(59b)} \\[2mm]
\mathrm{d}\widehat{Y}_t = \left(\frac{1}{2}\|\widehat{Z}_t\|^2 + \widehat{Z}_t^{\mathsf{T}}Z_t + \nabla\cdot f_t^- + \mathcal{G}_t\right)\mathrm{d}t + \widehat{Z}_t^{\mathsf{T}}\, \mathrm{d}W_t & \text{(59c)}
\end{cases}
$$

where we define $f_t^- := -f_t + g_t \widehat{Z}_t$ and $\mathcal{G}_t := {}^1\!/_2 \nabla^2 \cdot (g g_t^{\mathsf{T}})$. The nonlinear FK transformation reads

$$
\begin{aligned}
Y_t &= \log \Psi(t, X_t), & Z_t &= g(t, X_t)^{\mathsf{T}} \nabla \log \Psi(t, X_t), \\
\widehat{Y}_t &= \log \widehat{\Psi}(t, X_t), & \widehat{Z}_t &= g(t, X_t)^{\mathsf{T}} \nabla \log \widehat{\Psi}(t, X_t),
\end{aligned}
$$

which immediately suggests that

$$
\mathbb{E}[Y_t | X_t = x] = \log \Psi(t, x), \qquad \mathbb{E}[\widehat{Y}_t | X_t = x] = \log \widehat{\Psi}(t, x). \tag{60}
$$

Since SB-FBSDE was primarily developed in the context of generative modeling (Song et al., 2021), its training relies on computing the log-likelihood at the boundaries. These log-likelihoods can be obtained by noticing that $\log \rho(x, t) = \mathbb{E}[Y_t + \widehat{Y}_t | X_t = x]$, as implied by the Hopf-cole transform and Eq. (60). The above arguments also holds for the timeline of $s$ with equivalent $\widehat{\Psi}$ and $\Psi$ change their roles, vice versa.

## B.2 ON TIME DERIVATIVES OF KINETIC SYSTEMS

On second-order dynamic control of a kinetic system,[1] we can substitutes its time derivative $\partial_t f$ with $\left(\nabla_q f^{\mathsf{T}} \dot{q} + \nabla_{\dot{q}} f^{\mathsf{T}} \ddot{q}\right)$ where the control $x_t$ affects $q_t$ using arbitrary Hamiltonian, *e.g.*, affine-control (Lin et al., 2021; Liu et al., 2022a) .

**Lemma B.3.** *Suppose $h \in C^{1,2}$ and $\hbar \in C^{1,1,2}$ where $h(t, x(t)) = \hbar(q_t, \dot{q}_t, x_t)$ for $t \in [0, +\infty)$. Let both functions satisfy growth and Lipchitz conditions. Then, the time derivative of $h$ is drawn as*

$$
\frac{\partial h}{\partial t}(t, x(t)) = \left[\frac{\partial \hbar}{\partial q} + \frac{\partial \hbar}{\partial \dot{q}}\right](q_t, \dot{q}_t, x_t), \tag{61}
$$

*and $\nabla h$ and $\mathrm{Tr}(\mathrm{Hess}\, h)$ corresponds to $\nabla_{x_t} \hbar$ and $\mathrm{Tr}(\mathrm{Hess}_{x_t}\, \hbar)$ respectively with every $t$.*

*Proof.* Given standard drift and diffusion functions $f$ and $g$ recall the Itô lemma,

$$
\frac{\partial h}{\partial t} + \mathcal{A}_t h = \frac{\partial h}{\partial t} + \nabla h^{\mathsf{T}} f + \frac{1}{2} \mathrm{Tr}(g g^{\mathsf{T}} \mathrm{Hess}\, h) \tag{62}
$$

and the infinitesimal generation $\mathcal{A}_t$ depends on current $(q_t, \dot{q}_t, x_t)$. Now let us define stochastic process in combined coordinate $\mathrm{d}(q_t, \dot{q}_t, x_t) = \mu_t\, \mathrm{d}t + \sigma_t \mathrm{d}W_t$ well defined, *i.e.*,

$$
\mu_t = \begin{bmatrix} \dot{q}_t \\ \ddot{q}_t \\ f_t \end{bmatrix} \quad \text{and} \quad \sigma_t = \begin{bmatrix} \mathbf{0}_d \\ \mathbf{0}_d \\ g_t \end{bmatrix}.
$$

It is evident the infinitesimal generation of $\hbar$ and $h$ matched since both process exhibits $\mathcal{F}^{(k)}$-measurable Markovian kernel. Therefore, the time input incorporated for inputs of $\hbar$ the Itô formula for $\frac{\partial \hbar}{\partial t} = 0$ is expressed as

$$
\nabla h + \frac{1}{2}\sigma^2 \Delta h = \frac{\partial \hbar}{\partial q} + \frac{\partial \hbar}{\partial \dot{q}} + \frac{1}{2} g g^{\mathsf{T}} \mathrm{Tr}(\mathrm{Hess}_\xi\, \hbar) \tag{63}
$$

---

[1]This naturally extends to any settings with $k$-th order Markovian properties ($k \in \mathbb{N}$).

Equating algebraic expressions Eqs. (62) and (63),

$$\frac{\partial h}{\partial t} = \frac{\partial \hbar}{\partial q_t} + \frac{\partial \hbar}{\partial \dot{q}_t}, \quad \nabla h = \mathrm{Tr}(\mathrm{Hess}\,\hbar), \text{ and } \nabla \hbar = \mathrm{Tr}(\mathrm{Hess}\,\hbar), \tag{64}$$

for arbitrary $f$ and $g$. This concludes the proof. $\qquad \square$

This technique using the chain-rule is useful for solving the PDE in model-free manners.

### B.3 Derivations for Multi-Variate FBSDE

#### B.3.1 A Hopf-Cole Transformation

For an Hamiltonian $\mathcal{H}$, the general dynamic formulation is described with the following PDE:

$$\begin{cases} -\partial_t u + \mathcal{H}(x, \nabla u) - \frac{1}{2}\mathrm{Tr}(gg^\mathsf{T}\nabla^2 u) = 0, \\ \partial_t \rho - \nabla \cdot (\nabla_p \mathcal{H}(x, \nabla u)\,\rho) - \frac{1}{2}\nabla^2 \cdot (gg^\mathsf{T}\rho) = 0. \end{cases} \tag{65}$$

This can be viewed as blending of two optimality condition of PDEs; the first row indicates the HJE and the second row indicates Fokker–Planck (FP) equation (Buckdahn et al., 2017). We use the Hopf-Cole transform as follows

$$\Psi(t, x(t)) := \exp(-u(t, x(t))), \qquad \widehat{\Psi}(\xi(t), t) := \rho(t, x(t))\exp\big(u(t, x(t))\big).$$

This transform can be quite general since $\rho(t, x)$ is the density function which represents arbitrary measures. Multi-variate calculus yields

$$\nabla\Psi = -\exp(-u)\nabla u, \qquad \nabla^2\Psi = \exp(-u)[\nabla u \nabla u^\mathsf{T} - \nabla^2 u],$$

$$\nabla\widehat{\Psi} = \exp(u)(\rho\nabla u + \nabla\rho), \nabla^2\widehat{\Psi} = \exp(u)\Big[\rho\nabla u \nabla u^\mathsf{T} + \nabla\rho\nabla u^\mathsf{T} + \nabla u \nabla\rho^\mathsf{T} + \nabla^2\rho + \rho\nabla^2 u\Big].$$

Hence, we can draw the following derivations regarding the control-affine Hamiltonian, *i.e.*, $\mathcal{H}(x, \nabla u) = \frac{1}{2}\|g^\mathsf{T}\nabla u\|^2 - \nabla u^\mathsf{T} f$:

$$\frac{\partial\Psi}{\partial t} = \exp(-u)\left(-\frac{\partial u}{\partial t}\right)$$

$$= \exp(-u)\left(-\frac{1}{2}\|g^\mathsf{T}\nabla u\|^2 + \nabla u^\mathsf{T} f + \frac{1}{2}\mathrm{Tr}(gg^\mathsf{T}\nabla^2 u)\right)$$

$$= -\frac{1}{2}\mathrm{Tr}(gg^\mathsf{T}\nabla^2\Psi) - \nabla\Psi^\mathsf{T} f + \lambda\Psi\log\Psi$$

$$\frac{\partial\widehat{\Psi}}{\partial t} = \exp(u)\left(\frac{\partial\rho}{\partial t} + \rho\frac{\partial u}{\partial t}\right)$$

$$= \exp(u)\left(\left(\nabla\cdot(\rho(gg^\mathsf{T}\nabla u - f)) + \frac{1}{2}\nabla^2\cdot(gg^\mathsf{T}\rho)\right) + \rho\left(\frac{1}{2}\|g^\mathsf{T}\nabla u\|^2 - \nabla u^\mathsf{T} f - \frac{1}{2}\mathrm{Tr}(gg^\mathsf{T}\nabla^2 u)\right)\right)$$

$$= \frac{1}{2}\mathrm{Tr}(gg^\mathsf{T}\nabla^2\widehat{\Psi}) + \nabla\cdot(gg^\mathsf{T})^\mathsf{T}\nabla\widehat{\Psi} + \frac{1}{2}\nabla^2\cdot(gg^\mathsf{T})\widehat{\Psi} - \nabla\widehat{\Psi}^\mathsf{T} f - \widehat{\Psi}\nabla\cdot f$$

$$= \frac{1}{2}\nabla^2\cdot(gg^\mathsf{T}\widehat{\Psi}) - \nabla\widehat{\Psi}^\mathsf{T} f - \widehat{\Psi}\nabla\cdot f$$

$$= -\nabla\cdot(\widehat{\Psi} f)$$

and we have derivations for the control PDE (10).

#### B.3.2 Nonlinear Multi-variate Feynman-Kac Derivations

Let us apply the Itô's lemma to the $u := \log\Psi(t, X_t)$ where $X_t$ follows the forward equation:

$$\mathrm{d}\log\Psi = \frac{\partial\log\Psi}{\partial t} + \left[\nabla\log\Psi^\mathsf{T}\big(f + gg^\mathsf{T}\nabla\log\Psi\big) + \frac{1}{2}\mathrm{Tr}\big(gg^\mathsf{T}\nabla^2\log\Psi\big)\right]\mathrm{d}t + g\nabla\log\Psi^\mathsf{T}\mathrm{d}W_t.$$

Notice that the PDE of $\frac{\partial \log \Psi}{\partial t}$ is achieved by applying the Hopf-Cole transform

$$\frac{\partial \log \Psi}{\partial t} = \frac{1}{\Psi}\left(-\nabla \Psi^{\mathsf{T}} f - \frac{1}{2}\operatorname{Tr}(gg^{\mathsf{T}}\nabla^2 \Psi)\right)$$

$$= -\nabla \log \Psi^{\mathsf{T}} f - \frac{1}{2}\operatorname{Tr}(gg^{\mathsf{T}}\nabla^2 \log \Psi) - \frac{1}{2}\|g^{\mathsf{T}}\nabla \log \Psi\|^2.$$

Therefore, combining above differential terms yields

$$\mathrm{d}\log \Psi = \frac{1}{2}\|g\nabla \log \Psi\|^2\,\mathrm{d}t + g\nabla \log \Psi^{\mathsf{T}}\,\mathrm{d}W_t. \tag{66}$$

Also, apply the Itô lemma by instead substituting $u := \log \widehat{\Psi}(t, X_t)$

$$\mathrm{d}\log \widehat{\Psi} = \frac{\partial \log \widehat{\Psi}}{\partial t}\mathrm{d}t + \left[\nabla \log \widehat{\Psi}^{\mathsf{T}}(f + gg^{\mathsf{T}}\nabla \log \Psi) + \frac{1}{2}\operatorname{Tr}(gg^{\mathsf{T}}\nabla^2 \log \widehat{\Psi})\right]\mathrm{d}t + g\nabla \log \widehat{\Psi}^{\mathsf{T}}\mathrm{d}W_t.$$

Notice that the PDE of $\frac{\partial \log \widehat{\Psi}}{\partial t}$ obeys

$$\frac{\partial \log \widehat{\Psi}}{\partial t} = \frac{1}{\widehat{\Psi}}\left(-\nabla \cdot (\widehat{\Psi} f) + \frac{1}{2}\nabla^2 \cdot (gg^{\mathsf{T}}\widehat{\Psi})\right)$$

$$= -\nabla \log \Psi^{\mathsf{T}} f - \nabla \cdot f + \frac{1}{2}\|g^{\mathsf{T}}\nabla \log \widehat{\Psi}\|^2 + \frac{1}{2}\operatorname{Tr}(gg^{\mathsf{T}}\nabla^2 \log \widehat{\Psi}) + \frac{1}{2}\nabla^2 \cdot (gg^{\mathsf{T}})$$

where $\frac{1}{2}\nabla^2 \cdot (gg^{\mathsf{T}})$ is the adjustment term for non-constant diffusion $g$. This yields

$$\mathrm{d}\log \widehat{\Psi} = \left[-\nabla \cdot f + \frac{1}{2}\|g^{\mathsf{T}}\nabla \log \widehat{\Psi}\|^2 + g^{\mathsf{T}}\nabla \log \widehat{\Psi}^{\mathsf{T}}\nabla \log \Psi + \operatorname{Tr}(gg^{\mathsf{T}}\nabla^2 \log \widehat{\Psi})\right]\mathrm{d}t + g\nabla \log \widehat{\Psi}^{\mathsf{T}}\mathrm{d}W_t$$

$$= \left[\frac{1}{2}\|g^{\mathsf{T}}\nabla \log \widehat{\Psi}\|^2 + (g^{\mathsf{T}}\nabla \log \widehat{\Psi})^{\mathsf{T}}(g^{\mathsf{T}}\nabla \log \Psi) + \nabla \cdot (gg^{\mathsf{T}}\nabla \log \widehat{\Psi} - f) + \frac{1}{2}\nabla^2 \cdot (gg^{\mathsf{T}})\right]\mathrm{d}t + g\nabla \log \widehat{\Psi}^{\mathsf{T}}\mathrm{d}W_t,$$

Therefore, with the nonlinear FK transformation ([Pereira et al., 2020](#)), *i.e.*,

$$\begin{aligned}
Y_t &\equiv Y(t, X_t) = \log \Psi(t, X_t), & Z_t &\equiv Z(t, X_t) = g^{\mathsf{T}}\nabla \log \Psi(t, X_t) \\
\widehat{Y}_t &\equiv \widehat{Y}(t, X_t) = \log \widehat{\Psi}(t, X_t), & \widehat{Z}_t &\equiv \widehat{Z}(t, X_t) = g^{\mathsf{T}}\nabla \log \widehat{\Psi}(t, X_t)
\end{aligned} \tag{67}$$

we can write the SB-FBSDE of the system system

$$\mathrm{d}X_t = (f_t + gZ_t)\,\mathrm{d}t + g\,\mathrm{d}W_t$$

$$\mathrm{d}Y_t = \frac{1}{2}\|Z_t\|^2\,\mathrm{d}t + Z_t^{\mathsf{T}}\mathrm{d}W_t$$

$$\mathrm{d}\widehat{Y}_t = \left[\frac{1}{2}\|\widehat{Z}_t\|^2 + \widehat{Z}_t^{\mathsf{T}}Z_t + \nabla \cdot (g_t\widehat{Z}_t - f_t) + \frac{1}{2}\nabla^2 \cdot (gg_t^{\mathsf{T}})\right]\mathrm{d}t + \widehat{Z}_t^{\mathsf{T}}\mathrm{d}W_t$$

where we find forward control and reverse drift has the relationship with FK transformation as

$$f_t^+ = f_t + g_t Z_t \quad \text{and} \quad f_t^- = g_t\widehat{Z}_t - f_t$$

Derivation of the second FBSDEs system in time-reversal follows the identical flow, except that we need to rebase the PDE to the "reversed" time coordinate $s := T - t$. This can be also done by reformulating the HJE and FP equations under the $s$ coordinate, then applying the following Hopf-Cole transform:

$$\widehat{\Psi}(s, x) := \exp(-u(s, x)), \qquad \Phi(s, x) := \rho(s, x)\exp(u(s, x)). \tag{68}$$

Notice that we flip the role of $\widehat{\Psi}$ and $\Psi$ of the timeline $t$ and now relates to $s$ coordinate. Omitting the computation similar to Appendix B.3.1, we arrive at the following:

$$\begin{cases}
\frac{\partial \widehat{\Psi}(x,s)}{\partial s} = \nabla \widehat{\Psi} f - \frac{1}{2}g^2 \Delta \widehat{\Psi} \\
\frac{\partial \Psi(x,s)}{\partial s} = \nabla \cdot (\Psi f) + \frac{1}{2}g^2 \Delta \Psi
\end{cases} \quad s.t. \quad \begin{aligned} \widehat{\Psi}(\cdot, 0)\Psi(\cdot, 0) &= \nu \\ \widehat{\Psi}(\cdot, T)\Psi(\cdot, T) &= \mu \end{aligned} \tag{69}$$

Apply to Itô's lemma to $u := \log \widehat{\Psi}(s, \bar{X}_s)$ where $\bar{X}_s$ evolves along the reversed SDE:

$$\mathrm{d} \log \widehat{\Psi} = \frac{\partial \log \widehat{\Psi}}{\partial s} \mathrm{d}s + \left[ \nabla \log \Psi^{\mathsf{T}} \left( f + gg^{\mathsf{T}} \nabla \log \Psi \right) + \frac{1}{2} \mathrm{Tr} \left( gg^{\mathsf{T}} \nabla^2 \log \Psi \right) \right] \mathrm{d}s + g \nabla \log \Psi^{\mathsf{T}} \mathrm{d}W_s. \tag{70}$$

and notice that the PDE of $\frac{\partial \log \Psi}{\partial s}$ now obeys

$$\begin{aligned} \frac{\partial \log \widehat{\Psi}}{\partial s} &= \frac{1}{\widehat{\Psi}} \left( -\nabla \widehat{\Psi}^{\mathsf{T}} f - \frac{1}{2} \mathrm{Tr}(gg^{\mathsf{T}} \nabla^2 \Psi) \right) \\ &= -\nabla \log \widehat{\Psi}^{\mathsf{T}} f - \frac{1}{2} \mathrm{Tr}(gg^{\mathsf{T}} \nabla^2 \log \widehat{\Psi}) - \frac{1}{2} \| g^{\mathsf{T}} \nabla \log \widehat{\Psi} \|^2. \end{aligned}$$

This yields

$$\mathrm{d} \log \widehat{\Psi} = \frac{1}{2} \| g \nabla \log \widehat{\Psi} \|^2 \mathrm{d}s + g \nabla \log \widehat{\Psi}^{\mathsf{T}} \mathrm{d}W_s. \tag{71}$$

Similarly, apply the Itô's lemma to $u := \log \widehat{\Psi}(\bar{X}_s, s)$ where $\bar{X}_s$ follows the reversed SDE.

$$\mathrm{d} \log \Psi = \frac{\partial \log \Psi}{\partial s} \mathrm{d}s + \left[ \nabla \log \widehat{\Psi}^{\mathsf{T}} (f + gg^{\mathsf{T}} \nabla \log \Psi) + \frac{1}{2} \mathrm{Tr}(gg^{\mathsf{T}} \nabla^2 \log \widehat{\Psi}) \right] \mathrm{d}t + g \nabla \log \widehat{\Psi}^{\mathsf{T}} \mathrm{d}W_t. \tag{72}$$

and using the PDE of $\frac{\partial \log \widehat{\Psi}}{\partial s}$ yields

$$\begin{aligned} \mathrm{d} \log \Psi &= \left[ -\nabla \cdot f + \frac{1}{2} \| g^{\mathsf{T}} \nabla \log \Psi \|^2 + g^{\mathsf{T}} \nabla \log \Psi^{\mathsf{T}} \nabla \log \widehat{\Psi} + \mathrm{Tr}(gg^{\mathsf{T}} \nabla^2 \log \Psi) \right] \mathrm{d}t + g \nabla \log \Psi^{\mathsf{T}} \mathrm{d}W_t \\ &= \left[ \frac{1}{2} \| g^{\mathsf{T}} \nabla \log \Psi \|^2 + (g^{\mathsf{T}} \nabla \log \Psi)^{\mathsf{T}} (g^{\mathsf{T}} \nabla \log \widehat{\Psi}) + \nabla \cdot (gg^{\mathsf{T}} \nabla \log \Psi - f) + \frac{1}{2} \nabla^2 \cdot (gg^{\mathsf{T}}) \right] \mathrm{d}t + g \nabla \log \Psi^{\mathsf{T}} \mathrm{d}W_t, \end{aligned}$$

Therefore, with a nonlinear FK transformation

$$\begin{aligned} Y_s &\equiv Y(\bar{X}_s, s) = \log \Psi(\bar{X}_s, s), & Z_s &\equiv Z(\bar{X}_s, s) = g \nabla \log \Psi(\bar{X}_s, s), \\ \widehat{Y}_s &\equiv \widehat{Y}(\bar{X}_s, s) = \log \widehat{\Psi}(\bar{X}_s, s), & \widehat{Z}_s &\equiv \widehat{Z}(\bar{X}_s, s) = g \nabla \log \widehat{\Psi}(\bar{X}_s, s), \end{aligned}$$

we can rewrite the second FBSDE

$$\begin{aligned} \mathrm{d}\bar{X}_s &= (-f_s + g\widehat{Z}_t) \, \mathrm{d}s + g \mathrm{d}W_s \\ \mathrm{d}Y_s &= \left[ \frac{1}{2} \| Z_s \|^2 + Z_s^{\mathsf{T}} \widehat{Z}_s + \nabla \cdot (g_s Z_s + f_s) + Z_s^{\mathsf{T}} + \frac{1}{2} \nabla^2 \cdot (gg_s^{\mathsf{T}}) \right] \mathrm{d}W_s \\ \mathrm{d}\widehat{Y}_s &= \frac{1}{2} \| Z_s \|^2 + \widehat{Z}_t^{\mathsf{T}} \mathrm{d}W_s \end{aligned}$$

where we find the relationship $f_s^+ = f_s + g_s Z_s$ and $f_s^- = g_s \widehat{Z}_s - f_s$.

Finally, we present the SB-FBSDE for $X(t)$

$$\text{SB-FBSDE} \atop \text{w.r.t. (9a)} : \begin{cases} \mathrm{d}X_t = f_t^+ \mathrm{d}t + g_t \, \mathrm{d}W_t, & \text{(73a)} \\ \mathrm{d}Y_t = \frac{1}{2} \| Z_t \|^2 \mathrm{d}t + Z_t^{\mathsf{T}} \, \mathrm{d}W_t, & \text{(73b)} \\ \mathrm{d}\widehat{Y}_t = \left( \frac{1}{2} \| \widehat{Z}_t \|^2 + \widehat{Z}_t^{\mathsf{T}} Z_t + \nabla \cdot f_t^- + \mathcal{G}_t \right) \mathrm{d}t + \widehat{Z}_t^{\mathsf{T}} \, \mathrm{d}W_t, & \text{(73c)} \end{cases}$$

where $\mathcal{G}_t := 1/2 \nabla^2 \cdot (gg_t^{\mathsf{T}})$. Also, the SB-FBSDE for the $s$ time coordinate writes

$$\text{SB-FBSDE} \atop \text{w.r.t. (9b)} : \begin{cases} \mathrm{d}\bar{X}_s = f_s^- \mathrm{d}s + g \, \mathrm{d}W_s, & \text{(74a)} \\ \mathrm{d}Y_s = \left( \frac{1}{2} \| Z_s \|^2 + Z_s^{\mathsf{T}} \widehat{Z}_s + \nabla \cdot f_s^+ + \mathcal{G}_s \right) \mathrm{d}s + Z_s^{\mathsf{T}} \, \mathrm{d}W_s, & \text{(74b)} \\ \mathrm{d}\widehat{Y}_s = \frac{1}{2} \| \widehat{Z}_s \|^2 \mathrm{d}t + \widehat{Z}_s^{\mathsf{T}} \, \mathrm{d}W_s. & \text{(74c)} \end{cases}$$

The FBSDEs comprise a system of evolution toward a unique solution with respect to the cost of $1/2\|Z\|^2$ and $1/2\|\widehat{Z}\|^2$. Since $(Z, \widehat{Z})$ models the pure commitment of control, learning through Eq. (73b) and Eq. (74c) corresponds to satisfying Hamilton-Jacobi solution of the minimum control. Akin to temporal difference learning (Sutton & Barto, 2018) methods, Liu et al. (2022a) proposed a multi-step TD method via backward stochastic integration of the BSDEs (73c) and (73b) which iteratively elevates a variational lower bound of divergence.

## B.4 Forward and Reverse HJE loss

In SSBM, the drift loss in Eq. (16) is possible without the base drift function $f$. Instead, we can utilize another HJE loss function:

$$\mathcal{L}_{\text{HJE}}^+(\ell) = \mathbb{E}_{t,x}\left| \frac{\partial Y_\ell}{\partial t} + f_\ell^+ \cdot \nabla Y_\ell + \frac{1}{2}\operatorname{Tr}(gg^\mathsf{T}\nabla^2 Y_\ell) - \frac{1}{2}\left\|\nabla Y_\ell\right\|_{gg^\mathsf{T}}^2 \right| \tag{75}$$

to satisfy the equivalent learning with Eq. (16a), where $\mathcal{L}_{\text{HJE}}^-$ is symmetrically defined with $\widehat{Y}_\ell$. First, we can denote the forward and backward SDEs at iteration $\ell$ by

$$\mathrm{d}X_t^{(\ell)} = f_\ell^+\left(t, X_t^{(\ell)}\right)\mathrm{d}t + g\left(t, X_t^{(\ell)}\right)\mathrm{d}W_t, \quad X_0^{(\ell)} \sim \mu \tag{76a}$$

$$\mathrm{d}\bar{X}_s^{(\ell)} = f_\ell^-\left(s, \bar{X}_s^{(\ell)}\right)\mathrm{d}s + g\left(s, \bar{X}_s^{(\ell)}\right)\mathrm{d}W_s \quad \bar{X}_0^{(\ell)} \sim \nu \tag{76b}$$

The the probability density of random variable $X_t^{(\ell)}$ is represented with $\rho_\ell^+$ and that of $\bar{X}_s^{(\ell)}$ with $\rho_\ell^-$.

### B.4.1 Preliminary

**Lemma B.4.** *The following equality holds at any point $x \in \mathbb{R}^n$ such that $p(x) \neq 0$.*

$$\frac{1}{p(x)}\Delta p(x) = \|\nabla \log p(x)\|^2 + \Delta \log p(x)$$

*where $\Delta$ denotes the Laplacian operator.*

*Proof.* This can be proved by the log-nabla trick $\frac{1}{p(x)}\Delta p(x) = \frac{1}{p(x)}\nabla \cdot \nabla p(x) = \frac{1}{p(x)} \cdot (p(x)\nabla \log p(x))$. Applying the chain rule to the divergence $\nabla\cdot$ yields the desired result. $\square$

**Lemma B.5** (Proposition 1, Sect. 6.3.1, Vargas, 2021)**.** *In Eq. (76), denote respective densities $\rho_\ell^-$ and $\rho_\ell^+$. The following equation holds:*

$$\mathrm{d}\log\rho_\ell^- = \left[\nabla\cdot f^- + g\left(Z_\ell + \widehat{Z}_\ell\right)^\mathsf{T}\nabla\log\rho_\ell^- - \frac{1}{2}\|g\nabla\log\rho_\ell^-\|^2\right]\mathrm{d}t + (g\nabla\log\rho_\ell^-)^\mathsf{T}\mathrm{d}W_t.$$

*Proof.* Invoking Itô lemma with respect to the parameterized forward SDE (76a),

$$\mathrm{d}\log\rho_\ell^- = \left[\frac{\partial\log\rho_\ell^-}{\partial t} + (\nabla\log\rho_\ell^-)^\mathsf{T}f_\ell^+ + \frac{1}{2}\operatorname{Tr}[gg^\mathsf{T}\nabla^2\log\rho_\ell^-]\right]$$

where $\frac{\partial\log\rho_\ell^-}{\partial t}$ obeys

$$-\frac{\partial\rho_\ell^-}{\partial t} = -\nabla\cdot\left(f_\ell^-\rho_\ell^-\right) + \frac{1}{2}gg^\mathsf{T}\nabla^2\rho_\ell^-$$

$$\implies -\frac{\partial\log\rho_\ell^-}{\partial t} = \nabla\cdot f_\ell^- + \left(g\widehat{Z}_\phi - f_t\right)^\mathsf{T}\nabla\log\rho_\ell^- - \frac{1}{2\rho_\ell^-}gg^\mathsf{T}\nabla^2\log\rho_\ell^-.$$

By using the Nelson's duality (Nelson, 2001), substituting the above relation yields the desired results. $\square$

**Proposition B.6** (Proposition 1, Sect. 6.3.1, Vargas, 2021)**.**

$$D_{\text{KL}}(\rho_\ell^+\|\rho_\ell^-) = \int_0^T \mathbb{E}_{\rho_\ell^+(t,\cdot)}\left[\frac{1}{2}\|\widehat{Z}_\ell + Z_\ell\|^2 + \nabla\cdot f_\ell^-\right](t,x)\,\mathrm{d}t + \mathbb{E}_{\rho_\ell^+(0,\cdot)}[\log\mu(x)] - \mathbb{E}_{\rho_\ell^+(T,\cdot)}[\log\nu(x)]$$

*Proof.* Recall that the parameterized backward SDE (76b) can be reversed (Song et al., 2021; Nelson, 2001) as

$$\mathrm{d}\bar{X}_t^{(\ell)} = \left( f_t - g\widehat{Z}_\ell\big(t, \bar{X}_t^{(\ell)}\big) + gg^\mathsf{T}\nabla \log \rho_\ell^-\big(t, \bar{X}_t^{(\ell)}\big) \right)\mathrm{d}t + g\,\mathrm{d}W_t.$$

Then, we have

$$
D_{\mathrm{KL}}(\rho_\ell^+ \| \rho_\ell^-)
$$
$$
= \int_0^T \mathbb{E}_{\rho_\ell^+}\left[ \frac{1}{2}\|\widehat{Z}_\ell + Z_\ell - g\nabla \log \rho_\ell^-\|^2 \right](t,x)\mathrm{d}t + D_{\mathrm{KL}}\big(\mu \big\| \rho_\ell^-(0,\cdot)\big)
$$
$$
= \int_0^T \mathbb{E}_{\rho_\ell^+}\left[ \frac{1}{2}\|\widehat{Z}_\ell + Z_\ell\|^2 - g(\widehat{Z}_\ell + Z_\ell)^\mathsf{T}\nabla \log \rho_\ell^- + \frac{1}{2}\|g\nabla \log \rho_\ell^-\|^2 \right]\mathrm{d}t + D_{\mathrm{KL}}\big(\mu \big\| \rho_\ell^-(0,\cdot)\big)
$$
$$
= \int_0^T \mathbb{E}_{\rho_\ell^+}\left[ \frac{1}{2}\|\widehat{Z}_\ell + Z_\ell\|^2 + \nabla \cdot \big(g\widehat{Z}_\ell - f_t\big) \right]\mathrm{d}t - \mathbb{E}_{\rho_\ell^-}\left[ \int_0^T \mathrm{d}\log \rho_\ell^- \right] + D_{\mathrm{KL}}\big(\mu \big\| \rho_\ell^-(0,\cdot)\big)
$$
$$
= \int_0^T \mathbb{E}_{\rho_\ell^+}\left[ \frac{1}{2}\|\widehat{Z}_\ell + Z_\ell\|^2 + \nabla \cdot \big(g\widehat{Z}_\ell - f_t\big) \right]\mathrm{d}t + \mathbb{E}_{\rho_\ell^+(0,\cdot)}[\log \mu(x)] - \mathbb{E}_{\rho_\ell^+(T,\cdot)}[\log \nu(x)]
$$

Therefore, the proof is complete. $\qquad\square$

## C  IMPLEMENTATION DETAILS

### C.1  NETWORK ARCHITECTURE

For simulation with nonlinear functions, we utilize the deep neural network architecture proposed by Liu et al. (2022a). The architecture is charecterized by following schema:

$$\mathtt{out} = \mathtt{out\_mod}(\mathtt{x\_mod}(x) + \mathtt{t\_mod}(\mathtt{emebdding}(t)))$$

where each modele is

$$\mathtt{t\_mod} = \mathtt{Linear} \to \mathtt{SiLU} \to \mathtt{Linear}$$
$$\mathtt{x\_mod} = \mathtt{Linear} \to \mathtt{SiLU} \to \mathtt{Linear} \to \mathtt{SiLU} \to \mathtt{Linear} \to \mathtt{SiLU} \to \mathtt{Linear}$$
$$\mathtt{out\_mod} = \mathtt{Linear} \to \mathtt{SiLU} \to \mathtt{Linear} \to \mathtt{SiLU} \to \mathtt{Linear}.$$

We use the sinusoidal embedding for $\mathtt{emebdding}(\cdot)$. For the physical control tasks, the architecture is essential identical, except $t$ is replaced with the positional and velocity vector $(q, \dot{q})$ and $\mathtt{emebdding}(\cdot)$ is set to the identity function. We set hidden dimension to 256 for high-dimensional Gaussian transportation, and 128 for the rest.

### C.2  DETAILS ON EXPERIMENTS

For all the experiments, we used AdamW optimizer with the learning rate of $10^{-3}$ with decay $10^{-6}$. We used Euler-Maruyama with the timestep $\Delta t = 0.01$. For the control experiments, we used Lemma B.3 to calculate the time derivatives and the $\mathcal{L}_{\mathrm{FSDE}}$ and $\mathcal{L}_{\mathrm{FSDE}}$ was replaced with $\mathcal{L}_{\mathrm{HJE}}^{\pm}$ defined in Eq. (75). For the marginal distribution $(\mu, \nu)$, We set $\mathcal{N}(\mathbf{0}_d, 0.2\mathbf{I}_d)$, and the rest of the dimension is the Dirac delta function. This setting is possible since all the randomness is induced from the filtration of stochastic control. The ground-truth second-order dynamical function was used for all the control simulations, where we set 4 seconds for the time limit.

