# OpenReview forum: "Symmetrized Schrödinger Bridge Matching"
_ICLR.cc/2024/Conference — ICLR 2024 Conference Withdrawn Submission_

### Official Review · Reviewer_EfaF · 2023-10-28

**Soundness:** 3 good
**Presentation:** 3 good
**Contribution:** 2 fair
**Rating:** 6
**Confidence:** 4

**Summary:**

The paper proposes a symmetric variant of Sinkhorn that enjoys monotonic improvements and provides thorough theoretical analysis. An algorithm called SSBM is then devised and evaluated on a few low-dimensional toy examples.

**Strengths:**

- The paper offers a strong theoretical analysis for symmetric Sinkhorn, which makes the proposed method appealing.

- Writing is generally clear. Fig 1, 2 & Table 1 are all helpful for reading. I also appreciate the notation tables in Appendix.

- The SB-FBSDE presented in Appendix B is intriguing and could potentially have independent value in other applications.

**Weaknesses:**

I have conflicting feelings about this work. On one hand, I find the idea of symmetric Sinkhorn and the theoretical analysis quite intriguing and exciting. On the other hand, I have concerns about the proposed algorithm, as it appears to bear resemblance to several prior works that have not been included in the experimental comparisons. I think further clarification is needed to better distinguish the proposed approach from prior methods. If there are similarities or connections with prior works, it's important to acknowledge them properly in the paper to provide a clear context for readers. This will help in understanding the novelty and contributions of the current work within the broader research landscape.

- L_CDM seems akin to the matching objective introduced in DSB, except perhaps the distribution $q(x_t)$, which remains unclear in terms of how it is sampled in practice. If $q$ is indeed sampled in a manner similar to DSB, it'll be better to use L_DSB rather than reinventing the wheel.

- (15) introduces a penalty for deviations from the PDE in (12). This type of loss has been employed in several previous works [1-3] for training SB, with the main distinction being that (15) includes a flow version. However, it remains unclear what advantage the flow version provides, especially since the proposed method still necessitates the parameterization of two drift networks rather than a single ODE vector field.

- Likewise, both (16) and the approach in [2] involve regression between policy and value networks, with the only difference being the inclusion of an additional base drift in (16). If the base drift is known before hands, I feel like subtracting it out of (16) should make the learning easier?

- Can the authors include [2] in the experiment? As the objective in [2] consist of three components that serve the same roles to (14, 15, 16). This comparison could provide useful insights into the relative strengths of the proposed method.

- The proposed method relies on access to the score function of the marginals, as indicated in (15). This limitation should be appropriately acknowledged.

Other minor comments:
- It's unclear what "dynamic training" means in the caption of Table 1 until I read through Sec 1. Some clarification will be helpful.
- typo in Sec 1 (page 2): "ialso"
- typo in (15): the last term should be $\hat Y$

[1] A Machine Learning Framework for Solving High-Dimensional Mean Field Game and Mean Field Control Problem
[2] Deep Generalized Schrödinger Bridge
[3] Transport, Variational Inference and Diffusions: with Applications to Annealed Flows and Schrodinger Bridges

**Questions:**

- Does the proposed algorithm (Alg 4) require learning 4 networks, i.e., f+, f-, Y, $\hat Y$?
- What's the exact definition of "j" in Alg 4? Is it defined as in the last paragraph in Sec 4.2?
- What're the Gaussian used in the second part of the experiments? It's not clear from App C.2. Can the authors provide other metrics in additional to TV?
- Table 2 suggests that SSBM is more effective at preserving the marginal than minimizing the KL divergences. Do the authors have any insights or intuitions about why this is the case? It may be helpful to explain how this observation arises, as it is not explicitly implied by the algorithm itself.
- In App C.2, why is L_FSDE replaced with (75)?

---

### Official Review · Reviewer_pujg · 2023-10-30

**Soundness:** 3 good
**Presentation:** 4 excellent
**Contribution:** 3 good
**Rating:** 3
**Confidence:** 4

**Summary:**

There are several step-by-step contributions in this paper that lead to a unique and novel approach for solving SBPs in an action matching/ flow matching styled type of objective, with the following properties:

1. Rather than alternating 2 half bridges / projections, the new symmetrized methodology performs these projections in parallel followed by a scaling step. Whilst it is unlikely that this offers a huge speed up it may offer a stabler / learning scheme
2.  An HJB type equation in terms of the current field which allows for learning the current field based formulation of SBPs proposed in this work.
3. Slight better contraction for the static symmetric Sinkhorn which promises faster convergence than IPF.

The authors then proceed to explore their newly proposed algorithm across a suite of toy experiments.


[1] Shi, Y., De Bortoli, V., Campbell, A. and Doucet, A., 2023. Diffusion Schr\" odinger Bridge Matching. arXiv preprint arXiv:2303.16852.
[2] Peluchetti, S., 2023. Diffusion Bridge Mixture Transports, Schr\" odinger Bridge Problems and Generative Modeling. arXiv preprint arXiv:2304.00917.
[3] Koshizuka, T. and Sato, I., 2022, September. Neural Lagrangian Schr\"{o} dinger Bridge: Diffusion Modeling for Population Dynamics. In The Eleventh International Conference on Learning Representations.
[4] Vargas, F., Thodoroff, P., Lamacraft, A. and Lawrence, N., 2021. Solving schrödinger bridges via maximum likelihood. Entropy, 23(9), p.1134.
[5] Koshizuka, T. and Sato, I., 2022, September. Neural Lagrangian Schr\"{o} dinger Bridge: Diffusion Modeling for Population Dynamics. In The Eleventh International Conference on Learning Representations.
[6] Pavon, M., Trigila, G. and Tabak, E.G., 2021. The Data‐Driven Schrödinger Bridge. Communications on Pure and Applied Mathematics, 74(7), pp.1545-1573.
[7] Vargas, F. and Nüsken, N., 2023. Transport, Variational Inference and Diffusions: with Applications to Annealed Flows and Schr\" odinger Bridges. arXiv preprint arXiv:2307.01050.

**Strengths:**

This paper is extremely well written with a beautiful presentation of the idea and neat figures. Props to the authors for taking the time and effort to produce such a well-written manuscript. Furthermore, the ideas themselves are very interesting and creative from a conceptual perspective and could offer promise in improving/understanding SBP numerics.

**Weaknesses:**

# Algorithm

1. Seems slower than IPF, we went from solving 2 nonlinear opt problems to now 4, the fact there is some level of concurrency in the opt problem is not really a huge gain unless one has a rather large cluster, overall this methodology brings in more compute, and whilst there is some nice theory lightly motivating the approach it is really not clear that it offers a huge advantage.
2. IPF (Algorithm 3) isnt quite Sinkhorn (bear with me !) Sinkhorn is an instance of IPF (and can be very closely connected to the static IPF), but in its most general path space / dynamic setting IPF isnt quite coordinate ascent on a dual and whilst recent work [7] re-derive/interpret IPF as a coordinate ascent algo its still not quite the Sinkhorn Dual setting as the objective there is not convex and depends on initialization, unlike the Sinkhorn dual. The purpose of this point is I think one should be careful when porting over a convergence rate/guarantee from Sinkhorn over to IPF without thought, the clear guarantees of IPF go back to Ruschendorf's original work and whilst one can leverage more modern ideas using Bregman projections and relate to Sinkhorn, I'd be quite careful in making strong claims here, specially when one performs approximate IPF projections. Similarly with Algorithm 4 it is more akin to IPF+matching than exactly symmetrised Sinkhorn, so some care should be taken when porting over claims. To give an example the proof for the convergence of IDBM matching [2] (a type of SB matching) is quite different and simpler to that of IPF and even less so related to your classical Sinkhron sketch.
3. Following from the previous point convergence guarantees for Algorithm 4 are not 100% clear from the manuscript, and it is ok if they are lacking, the motivation for the algorithm is clear, however, the authors should be more clear/precise in that in the current manuscript convergence guarantees are not provided.
4. The claim that this scheme provides more stable NN training is largely unverified both theoretically and empirically. The authors should not overclaim and instead stick to what has been shown, e.g. a faster convergence for the akin static Sinkhorn variant. The better/stabler NN training seems both misleading and unverified.
5. Detailed conceptual (and empirical) comparison to IDBM [2] should be done, as superficially has some of the programmatic advantages proposed in this work (e.g. no sequential projections).

# Experiments

The experiments are quite weak if the selling point is something like more stable training then the authors should really explore more challenging reference processes, in particular rare events arising from MD potentials, which to this date most SBP schemes fail to solve nicely even in 2D, for example:

1. The double well potential in  [4] or the various easier well-like potentials in [5]
2. The  Alanine Dipeptide potential in a TSP like setup see [3]  also the other potentials of the same paper  Polyproline Helix, Chignolin etc. The source and target distributions can be simple Gaussians centered at meta-stable states, code from this paper provide this.

In short the physical experiments provided are just a bit too artificial and if one is going to provide a method for stabler training it should really be explored with challenging MD/stat mech reference which actually gave to rise to the study of SBPs in the first place [6].

It seems that looking at Table 2 the results do seem quite incremental as we can see that SB-CFM completely outperforms the proposed approach across 2 metrics (relative entropies) and temporal TV actually is reached/surpassed by prior work a couple of times. Empirically I can't quite see the strong/case story for this approach with the current set of experiments.

Another point is I believe the authors should compare to IDBM [2] which also removes the sequential IPF-like projections and instead imposes both boundary constraints in one step.

# Typos

1. Prop 4.2:  Shouldn't Girsanov involve drifts in the same direction so $||f^- - \gamma^-||^2$ for example or $||f^+ - \gamma^+||^2$ rather than $||f^+ - \gamma^-||^2$ having cross drifts like this would either bring an extra score term or a $\nabla \cdot \gamma^-$ term.

**Questions:**

1. I am a bit confused with the nomenclature, what you call Schrodinger Bridge matching (e.g. Algorithm 3) really just is plain ol vanilla IPF, in practice you will note what De Bortoli et al 2021 or Vargas et al 2021 do is that they optimise forward KLs aided by ancestral sampling (effectively doing maximum likelihood) as opposed to reverse KLs as in Algorithm 3. I do believe De Bortoli et al 2021 at some point do use some terminology like mean matching formulas or something similar however Schrodinger Bridge matching is itself a completely different algorithm to IPF a bit more dual in nature see [1] or [2] for this method. In short at this point in the story there are now flow matching ideas so the terminology is very confusing and a bit incorrect. I think you have abstracted too much in algorithm 1 to the point where you have simply just written pseudocode for IPF and called it matching.

Also note, work such as [2] (IDBM) which is one of the concurrent papers on Schrodinger bridge matching does not perform the sequential projections that you have argued SB matching algorithms do via Algorithm 1 (which is just IPF). Methods such as IDBM at a high level have a similar computational setup but arguably perform less intermediate projections/subproblems than the proposed approach in this work.

2. What you call HJE  looks very akin to an HJB type equation, in fact, if one applies Hopf-Cole to the FK PDEs in Equation 10 which results in a coupled system of HJB equations and then combining them and subbing in v_t will very likely yield a very closely related equation to your HJE. I would still think of this as an HJB-type equation, is there a reason to drop Bellman? poor Bellman.

3. Why call the approach time-symmetric (sec 4.2) why not velocity/current field, or fluid mechanics based or something a bit more in line with the terminology in the Benamou-Brenier's paper,  time symmetry because the velocity is parametrized in terms of both forwards and backward drift distracted me as a reader, it would be ideal to explain why.  I can see it has the symmetry theme from the earlier sections but I can't fully grasp the name or time symmetrization terminology

4. Where you mention "... flow matching models (Liu et al., 2023; Shi et al., 2023)." please cite [2] as it is concurrent to both works and virtually the same algorithm as in Shi et al. 2023.

5. The objective in Eq 15 is very reminiscent of an action-matching styled objective, if so the authors should more carefully acknowledge this.

---

### Official Review · Reviewer_8RJL · 2023-10-31

**Soundness:** 2 fair
**Presentation:** 1 poor
**Contribution:** 2 fair
**Rating:** 3
**Confidence:** 4

**Summary:**

In this paper, the authors argue that the slow training of Schrodinger Bridges is due to their iterative nature. To solve this problem, they introduce different methods that allows to train backward and forward models jointly. Doing so they derive symmetrized version of existing algorithms. They start by introducing a symmetrized version of the Sinkhorn algorithm based on [1]. They prove that in the case of a bounded cost the algorithm converges exponentially fast. Then, they turn to the dynamic formulation of entropic OT and in particular leverage tools from Nelson duality [2]. They recall a flux equation which was notably described in [3]. The main interest of this evolution equation is that it is symmetric. Finally, they turn to the main methodological contribution of their paper which is an algorithm that trains jointly  the forward and the backward in a version of [4]. The paper is concluded with experiments in 2D, high dimensional Gaussian and 2D Physical Mass Control.

[1] Kurras (2015) -- Symmetric Iterative Proportional Fitting

[2] Nelson (2001) -- Dynamical theories of Brownian Motion

[3] Chen et al. (2017) -- Matrix optimal mass transport: a quantum mechanical approach.

[4] Shi et al. (2023) --Diffusion Schrödinger Bridge Matching

**Strengths:**

* The paper tackles an important problem in the Schrodinger Bridge litterature, namely the joint training of Schrodinger Bridge. This problem is of interest for the community and progress in that direction is interesting.

* The introduction of the symmetrized Sinkhorn algorithm and the relationships with [1] are very interesting. I think this is a promising area for future research.

* In low dimensions some of the results seem promising.

[1] Kurras (2015) -- Symmetric Iterative Proportional Fitting

**Weaknesses:**

* My main concern is with the writing of the paper. I found the paper to read more like a collection of statements than a proper, motivated, piece of research. I will give more precise examples in the next points but I think that significant rewriting is necessary in order for this paper to be considered for acceptance. As of now, too many details are missing and it is not clear what is the motivation of certain sections.

* Starting with the symmetrized Sinkhorn algorithm. It seems that a lot of the methodology borrows from [1]. It is necessary to compare with the algorithms proposed in [1]. What is the novelty here (if the algorithm is novel?)? If so, what are the new properties of this algorithm when compared to the ones proposed in [1]. How is this new symmetrized version of the Sinkhorn algorithm linked with the rest of the paper?

* The theoretical analysis has an interesting part (Proposition 3.1) when it comes to the linear convergence in the case of bounded cost. However, Proposition 3.2 only shows the monotonicity of the algorithm. The convergence of the algorithm in the general case is not addressed. Am I misunderstanding something? Is this a limitation of the paper? In any case more details should be given here to understand why Proposition 3.2 is showcased here.

* Is Section 4.2 "Time-symmetric approach to dynamical SB problem" a background section ? Or is there a contribution to be found here?

* My main concerns are with Section 4.3 which contains the algorithm. I could not understand the algorithm even after reading the appendices. I don't understand why the presented procedure makes sense. Is it related to the symmetrized Sinkhorn algorithm presented earlier in the paper? If so, how? What are the links with Section 4.2? How are the authors ensured of the convergence of the algorithm? It seems also that the authors confuse the loss used in flow matching type approaches [2,3] and the ones used in diffusion models like approaches [4,5]. It seems that the authors use a loss similar to [2,3], am I wrong?

* Minimizing (15) seems very unstable and not scalable as it requires differentiating the neural network to compute the loss. How is that handled numerically? Are the authors using a Hutchinson estimator? How does this loss affects the scalability of the algorithm?

* It also seems that the authors need to have access to the unnormalized densities to compute the boundary conditions. If this is the case this is a huge restriction on the class of problems that can be tackled with this approach. I spotted this limitation with the sentence following (15) so it is not clear to me how is the consistency enforced. The limitation is never discussed in the paper (in fact this is a more general remark, limitations of the algorithm are scarcely discussed in the paper).

Minor comments:

* Is a square missing in (16a) and (16b)?

* I don't understand the sentence following Proposition 3.1.

* Table 1 is difficult to read and understand

* Typo "convergence but ialso" in the Introduction

* [3] should also be cited when introducing [2] (works are concurrent)

[1] Kurras (2015) -- Symmetric Iterative Proportional Fitting

[2] Shi et al. (2023) -- Diffusion Schrödinger Bridge Matching

[3] Peluchetti (2023) -- Diffusion Bridge Mixture Transports, Schrödinger Bridge Problems and Generative Modeling

[4] De Bortoli et al. (2021) -- Diffusion Schrödinger Bridge with Applications to Score-Based Generative Modeling

[5] Vargas et al. (2021) -- Solving Schrödinger Bridges via Maximum Likelihood

**Questions:**

Please address the remarks listed under the "Weaknesses" section. I believe that a lot of emphasis should be put on explaining the contributions (and their limitations...) more clearly. As of now the paper reads like a collection of statements accompanied with an algorithm. Do the authors plan to release the code? I could not find any anonymized version of the code and it would be very useful to verify some of the results and comparisons made by the authors.

---

### Official Review · Reviewer_Ksvt · 2023-10-31

**Soundness:** 1 poor
**Presentation:** 1 poor
**Contribution:** 2 fair
**Rating:** 3
**Confidence:** 4

**Summary:**

The authors propose an additional training step in IPF training of diffusion schrodinger bridges in order to update both forward and backward drifts jointly. The proposed method is a mixture of IPF [1] and IMF [2].

[1] Bortoli et al Diffusion Schrodinger Bridge, 2021
[2] Shi et al Diffusion Schrodinger Bridge Matching, 2023

**Strengths:**

In principle using an IMF step with IPF should improve performance. It is interesting how one can leverage the probability flux to find a marginal preserving coupling then use bridge matching to find drifts from the coupling.

Empirical performance on simple Gaussians seems well performing relative to prior methods.

**Weaknesses:**

The paper is not written very clearly. This is perhaps an understatement, it is very difficult to read and understand what the authors propose and what the contributions are. Some notation are not properly defined, algorithms lack details and multiple losses not properly detailed, evaluation metrics not defined. Overall I believe this is far from being ready for publication.

It is not clear why one would use IPF and IMF steps together. What are the disadvantages of IMF?

In [3] appendices I believe there is a symmetric or joint update for both forward and backward drifts.

**Clarity**
- "We leave more algorithmic details in the appendix." - I cannot find details in the appendix
- " SBP relies on finite models" what are finite models?

**Experimental results**
It is not clear what metrics are being used for comparison. There is a lack of detail. Are these metrics for terminal measures only?

What is Forward Path Relative Entropy or Temporal Variation? Where are these defined?

What is being measured e.g. table 3, is this a comparison to the ground truth. Rectified flow does not solve the regularized OT problem so it is odd if this is close to the ground truth. If just testing generative performance then clearly non regularized diffusion models which are state of the art would perform best, and correspond to DSB with infinite regularization.


Minor:
" It has been verified that the SB is aligned with both score and flow matching (Liu et al., 2023; Shi et al., 2023)." Not sure what this means.

Claims need citation:
"For instance, SB generally performs score matching where the first training stage of IPF is equivalent to the exact score matching"
Should cite original SB paper for this, [1].

" SB with nonlinear networks with Sinkhorn algorithm (Var- gas et al., 2021; De Bortoli et al., 2021; Chen et al., 2022)"
Vargas uses GP not a network, so should not be cited here.

Error:
[2] is not a schrodinger bridge despite claims. It is performing bridge matching on a data driven coupling but bridge matching wrt brownian bridge. This induces a Markovian projection given the coupling from data does not correspond to the path being bridged. Details of this are given in [3].


[1] Bortoli et al Diffusion Schrodinger Bridge with application to score based generative modelling, 2021
[2] Liu et al I2sb: Image-to-image schro ̈dinger bridge., 2023
[3] Shi et al. Diffusion schr\” odinger bridge matching, 2023

**Questions:**

Why is it beneficial to use IPF updates over IMF? The only benefit I can see is for non linear reference diffusions.